# Tracking Without Re-recognition in Humans and Machines

**Drew Linsley**[*1]**, Girik Malik**[*2]**, Junkyung Kim**[3]**,**

**Lakshmi N Govindarajan**[1]**, Ennio Mingolla**[†2]**, Thomas Serre**[†1]
drew_linsley@brown.edu and malik.gi@northeastern.edu

## Abstract

Imagine trying to track one particular fruitfly in a swarm of hundreds. Higher biological visual systems have evolved to track moving objects by relying on both their appearance and their motion trajectories. We investigate if state-of-the-art spatiotemporal deep neural networks are capable of the same. For this, we introduce *PathTracker*, a synthetic visual challenge that asks human observers and machines to track a target object in the midst of identical-looking "distractor" objects. While humans effortlessly learn *PathTracker* and generalize to systematic variations in task design, deep networks struggle. To address this limitation, we identify and model circuit mechanisms in biological brains that are implicated in tracking objects based on motion cues. When instantiated as a recurrent network, our circuit model learns to solve *PathTracker* with a robust visual strategy that rivals human performance and explains a significant proportion of their decision-making on the challenge. We also show that the success of this circuit model extends to object tracking in natural videos. Adding it to a transformer-based architecture for object tracking builds tolerance to visual nuisances that affect object appearance, establishing the new state of the art on the large-scale TrackingNet challenge. Our work highlights the importance of understanding human vision to improve computer vision.

## 1 Introduction

Lettvin and colleagues [1] presciently noted, "The frog does not seem to see or, at any rate, is not concerned with the detail of stationary parts of the world around him. He will starve to death surrounded by food if it is not moving." Object tracking is fundamental to survival, and higher biological visual systems have evolved the capacity for two distinct and complementary strategies to do it. Consider Figure 1: can you track the object labeled by the yellow arrow from left-to-right? The task is trivial when appearance cues, like color, make it possible to solve the temporal correspondence problem by "re-recognizing" the target in each frame (Fig. 1a). However, this strategy is not effective when objects cannot be discriminated by their appearance alone (Fig. 1b). In this case integration of object motion over time is necessary for tracking. Humans are capable of tracking objects by their motion when appearance is uninformative [2, 3], but it is unclear if the current generation of neural networks for video analysis and tracking can do the same. To address this question we introduce *PathTracker*, a synthetic challenge for object tracking without re-recognition (Fig. 1c).

---

[*†]These authors contributed equally to this work.

[1]Carney Institute for Brain Science, Brown University, Providence, RI

[2]Northeastern University, Boston, MA

[3]DeepMind, London, UK

35th Conference on Neural Information Processing Systems (NeurIPS 2021).

Leading models for video analysis rely on object classification pre-training. This gives them access to rich semantic representations that have supported state-of-the-art performance on a host of tasks, from action recognition to object tracking [4–6]. As object classification models have improved, so too have the video analysis models that depend on them. This trend in model development has made it unclear if video analysis models are effective at learning tasks when appearance cues are uninformative. The importance of diverse visual strategies has been highlighted by synthetic challenges like *Pathfinder*,

a visual reasoning task that asks observers to trace long paths embedded in a static cluttered display [7, 8]. *Pathfinder* tests object segmentation when appearance cues like category or shape are missing. While humans can easily solve it [8], deep neural networks struggle, including state-of-the-art vision transformers [7–9]. Importantly, models that learn an appropriate visual strategy for *Pathfinder* also exhibit more efficient learning and improved generalization on object segmentation in natural images [10, 11]. Our *PathTracker* challenge extends this line of work into video by posing an object tracking problem where the target can be tracked by motion and spatiotemporal continuity, not category or appearance.

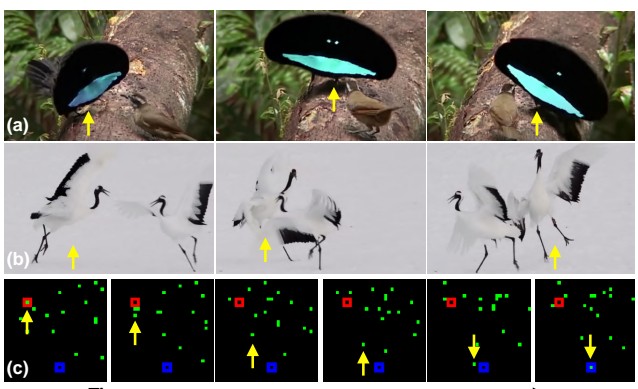

Figure 1: The appearance of objects makes them (*a*) easy or (*b*) hard to track. We introduce the *PathTracker* Challenge (*c*), which asks observers to track a particular green dot as it travels from the red-to-blue markers, testing object tracking when re-recognition is impossible.

**Contributions.** Humans effortlessly solve our novel *PathTracker* challenge. A variety of state-of-the-art models for object tracking and video analysis do not.

- We find that neural architectures including R3D [12] and state-of-the-art transformer-based TimeSformers [5] are strained by long *PathTracker* videos. Humans, on the other hand, are far more effective at solving these long *PathTracker* videos.

- We describe a solution to *PathTracker*: a recurrent network inspired by primate neural circuitry involved in object tracking, which renders decisions that are strongly correlated with humans.

- These same circuit mechanisms improve object tracking in natural videos through a motion-based strategy that builds tolerance to changes in target object appearance, resulting in the state-of-the-art score on TrackingNet [13].

- We release all *PathTracker* data, code, and human psychophysics at `http://bit.ly/InTcircuit` to spur interest in the challenge of tracking without re-recognition.

## 2   Related Work

**Models for video analysis**   A major leap in the performance of models for video analysis came from using networks which are pre-trained for object recognition on large image datasets [4]. The recently introduced TimeSformer [5] achieved state-of-the-art performance with weights initialized from an image categorization transformer (ViT; [14]) that was pre-trained on ImageNet-21K. The modeling trends are similar in object tracking [15], where successful models rely on "backbone" feature extraction networks trained on ImageNet or Microsoft COCO [16] for object recognition or segmentation [6, 17].

**Shortcut learning and synthetic datasets**   A byproduct of the great power of deep neural network architectures is their vulnerability to learning spurious correlations between inputs and labels. Perhaps because of this tendency, object classification models have trouble generalizing to novel contexts [18, 19], and render idiosyncratic decisions that are inconsistent with humans [20–22]. Synthetic datasets are effective at probing this vulnerability because they make it possible to control spurious image/label correlations – providing a fairer test of the computational abilities of these models. For example, the *Pathfinder* challenge was designed to test if neural architectures can trace long curves in clutter – a

visual computation associated with the earliest stages of visual processing in primates. That challenge identified diverging visual strategies between humans and transformers that are otherwise state of the art in natural image object recognition [9, 14]. Other challenges like Bongard-LOGO [23], cABC [8], SVRT [24], and PSVRT [25] have highlighted limitations of leading neural network architectures that would have been difficult to identify using natural image benchmarks like ImageNet [26]. These limitations have inspired algorithmic solutions based on neural circuits discussed in SI §1.

**Translating circuits for biological vision into artificial neural networks**    While the *Pathfinder* challenge [7] presents immense challenges for transformers and deep convolutional networks [8], it can be solved by a simple model of intrinsic connectivity in visual cortex, with orders-of-magnitude fewer parameters than standard models for image categorization. This model was developed by translating descriptive models of neural mechanisms from Neuroscience into an architecture that can be fit to data using gradient descent [7, 11]. Others have found success in modeling object tracking by drawing inspiration from "dual stream" theories of appearance and motion processing in visual cortex [27–30], or basing the architecture off of a partial connectome of the Drosophila visual system [31]. We adopt a similar approach in the current work, identifying mechanisms for object tracking without re-recognition in Neuroscience, and developing those into differentiable operations with parameters that can be optimized by gradient descent. This approach has the dual purpose of introducing task-relevant inductive biases into computer vision models, and developing theory on their relative utility for biological vision.

**Multi-object tracking in computer vision**    The classic psychological paradigms of multi-object tracking [2] motivated the application of models, like Kalman filters, which had tolerance to object occlusion when they relied on momentum models [32]. However, these models are computationally expensive, hand-tuned, and because of this, no longer commonly used in computer vision [33]. More recent approaches include flow tracking on graphs [34] and motion tracking models that are relatively computationally efficient [35, 36]. However, even current approaches to multi-object tracking are not learned, instead relying on extensive hand tuning [37, 38]. In contrast, the point of *PathTracker* is to understand the extent to which state-of-the-art neural networks are capable of tracking a single object in an array of distractors.

## 3   The *PathTracker* Challenge

**Overview**    *PathTracker* asks observers to decide whether or not a target dot reaches a goal location (Fig. 2). The target dot travels in the midst of a pre-specified number of distractors. All dots are identical, and the task is difficult because of this: (*i*) appearance is not useful for tracking the target, and (*ii*) the paths of the target and distractors can momentarily "cross" and occupy the same space, making it impossible to individuate them in that frame and meaning that observers cannot solely rely on the target's location to solve the task. This challenge is inspired by object tracking paradigms in cognitive psychology [2, 3, 39], which suggest that humans might tap into mechanisms for motion perception, attention and working memory to solve a task like *PathTracker*.

The trajectories of target and distractor dots are randomly generated, and the target occasionally crosses distractors (Fig. 2). These object trajectories are smooth by design, giving the appearance of objects meandering through a scene, and the difference between the coordinates of any dot on successive frames is no more than 2 pixels with less than $20°$ of angular displacement. In other words, dots never turn at acute angles. We develop different versions of *PathTracker* with varying degrees of complexity based on the number of distractors and/or the length of videos. These variables change the expected number of times that distractors cross the target and the amount of time that observers must track the target (Fig. 2). To make the task as visually simple as possible and maximize contrast between dots and markers, the dots, start, and goal markers are placed on different channels in $32\times32$ pixel three-channel images. Markers are stationary throughout each video and placed at random locations. Examples videos can be viewed at `http://bit.ly/InTcircuit`.

**Human benchmark**    We began by testing if humans can solve *PathTracker*. We recruited 180 individuals using Amazon Mechanical Turk to participate in this study. Participants viewed *PathTracker* videos and pressed a button on their keyboard to indicate if the target object or a distractor reached the goal. These videos were played in web browsers at $256\times256$ pixels using HTML5, which helped ensure consistent frame rates [40]. The experiment began with an 8-trial "training" stage, which

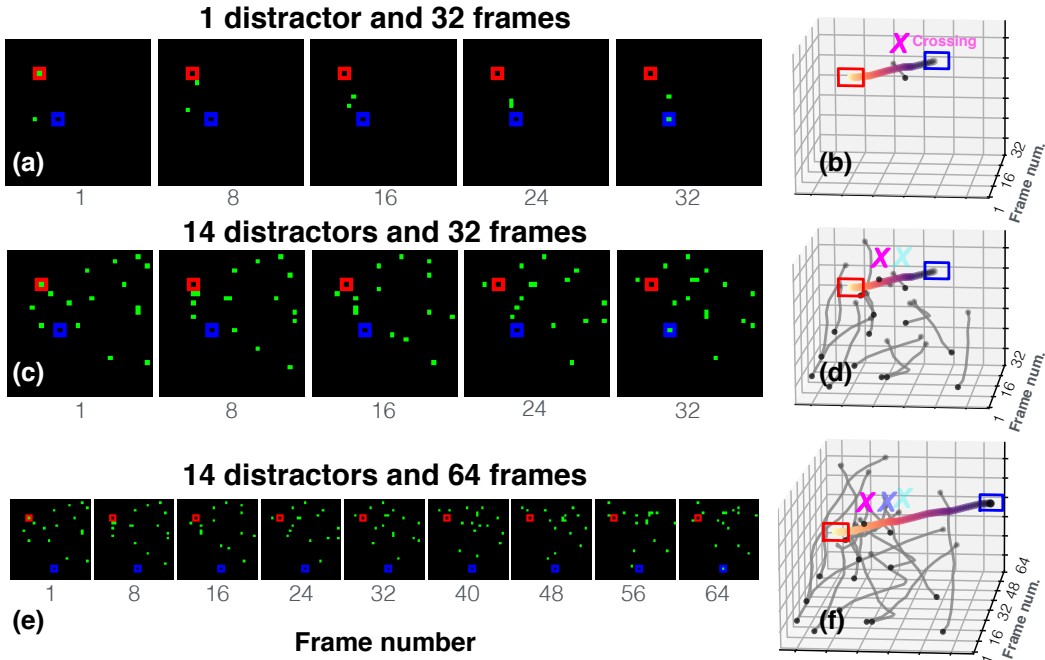

Figure 2: *PathTracker* is a synthetic visual challenge that asks observers to watch a video clip and answer if a target dot starting in a red marker travels to a blue marker. The target dot is surrounded by identical "distractor" dots, each of which travels in a randomly generated and curved path. In positive examples, the target dot's path ends in the blue square. In negative examples, a "distractor" dot ends in the blue square. The challenge of the task is due to the identical appearance of target and distractor dots, which, as we will show, makes appearance-based tracking strategies ineffective. Moreover, the target dot can momentarily occupy the same location as a distractor when they cross each other's paths, making it impossible to individuate them in that frame and compelling strategies like motion trajectory extrapolation or working memory to recover the target track. (*b*) A 3D visualization of the video in (*a*) depicts the trajectory of the target dot, traveling from red-to-blue markers. The target and distractor cross approximately half-way through the video. (*c,d*) We develop versions of *PathTracker* that test observers sensitivity to the number of distractors and length of videos (*e,f*). The number of distractors and video length interact to make it more likely for the target dot to cross a distractor in a video (compare the one X in *b* vs. two in *d* vs. three in *f*; see SI §2 for details).

familiarized participants with the goal of *PathTracker*. Next, participants were tested on 72 videos. The experiment was not paced and lasted approximately 25 minutes, and participants were paid $8 for their time. See `http://bit.ly/InTcircuit` and SI §2 for an example and more details.

Participants were randomly entered into one of two experiments. In the first experiment, they were trained on the 32 frame and 14 distractor *PathTracker*, and tested on 32 frame versions with 1, 14, or 25 distractors. In the second experiment, they were trained on the 64 frame and 14 distractor *PathTracker*, and tested on 64 frame versions with 1, 14, or 25 distractors. All participants viewed unique videos to maximize our sampling over the different versions of *PathTracker*. Participants were significantly above chance on all tested conditions of *PathTracker* ($p < 0.001$, test details in SI §2). They also exhibited a significant negative trend in performance on the 64 frame datasets as the number of distractors increased ($t = -2.74$, $p < 0.01$). There was no such trend on the 32 frame datasets, and average accuracy between the two datasets was not significantly different. These results validate our initial design assumptions: humans can solve *PathTracker*, and manipulating distractors and video length increases difficulty.

## 4   Solving the *PathTracker* challenge

Can leading models for video analysis match humans on *PathTracker*? To test this question we surveyed a variety of architectures that are the basis for leading approaches to many video analysis tasks, from object tracking to action classification. The models fall into four groups: (*i*) deep

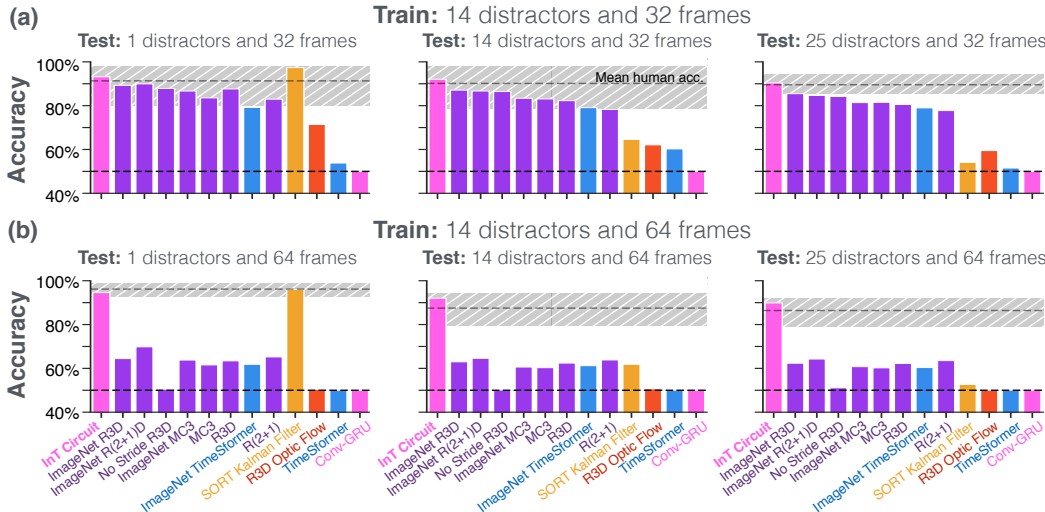

Figure 3: Model accuracy on the *PathTracker* challenge. Video analysis models were trained to solve 32 (*a*) and 64 frame (*b*) versions of challenge, which featured the target object and 14 identical distractors. Models were tested on *PathTracker* datasets with the same number of frames but 1, 14, or 25 distractors (left/middle/right). Colors indicate different instances of the same type of model. Grey hatched boxes denote 95% bootstrapped confidence intervals for humans. Only our InT Circuit rivaled humans on each dataset.

convolutional networks (CNNs), (*ii*) transformers, (*iii*) recurrent neural networks (RNNs), and (*iv*) Kalman filters. The deep convolutional networks include a 3D ResNet (R3D [12]), a space/time separated ResNet with "2D-spatial + 1D-temporal" convolutions (R(2+1)D [12]), and a ResNet with 3D convolutions in early residual blocks and 2D convolutions in later blocks (MC3 [12]). We trained versions of these models with random weight initializations and weights pretrained on ImageNet. We included an R3D trained from scratch without any downsampling, in case the small size of *PathTracker* videos caused learning problems (see SI §3 for details). We also trained a version of the R3D on optic flow encodings of *PathTracker* (SI §3). For transformers, we turned to the TimeSformer [5]. We evaluated two of its instances: (*i*) where attention is jointly computed for all locations across space and time in videos, and (*ii*) where temporal attention is applied before spatial attention, which results in massive computational savings. Both models performed similarly on *PathTracker*. We report the latter version here as it was marginally better (see SI §3 for performance of the other, joint space-time attention TimeSformer). We include a version of the TimeSformer trained from scratch, and a version pre-trained on ImageNet-20K. Note that state-of-the-art transformers for object tracking in natural videos feature similar deep and multi-headed designs [6]. For the RNNs, we include a convolutional-gated recurrent unit (Conv-GRU) [41]. Finally, our exemplar Kalman filter is the standard Simple and Online Realtime Tracking (SORT) algorithm, which was fed coordinates of the objects extracted from each frame of every video [37].

**Method** The visual simplicity of *PathTracker* cuts two ways: it makes it possible to compare human and model strategies for tracking without re-recognition as long as the task is not too easy. Prior synthetic challenges like *Pathfinder* constrain sample sizes for training to probe specific computations [7–9]. We adopted the following strategy to select a training set size that would help us test tracking strategies that do not depend on re-recognition. We took Inception 3D (I3D) networks [4], which have been a strong baseline architecture in video analysis over the past several years, and tested their ability to learn *PathTracker* as we adjusted the number of videos for training. As we discuss in SI §1, when this model was trained with 20K examples of the 32 frame and 14 distractor version of *PathTracker* it achieved good performance on the task without signs of overfitting. We therefore trained all models in subsequent experiments with 20K examples. By happenstance, this dataset size gives *PathTracker* a comparable number of frames to large-scale real world tracking challenges like LaSOT [42] and GOT-10K [43].

We measure the ability of models to learn *PathTracker* and systematically generalize to novel versions of the challenge when trained on 20K samples. We trained models using a similar approach as in our human psychophysics. Models were trained on one version of *PathTracker*, and tested on other

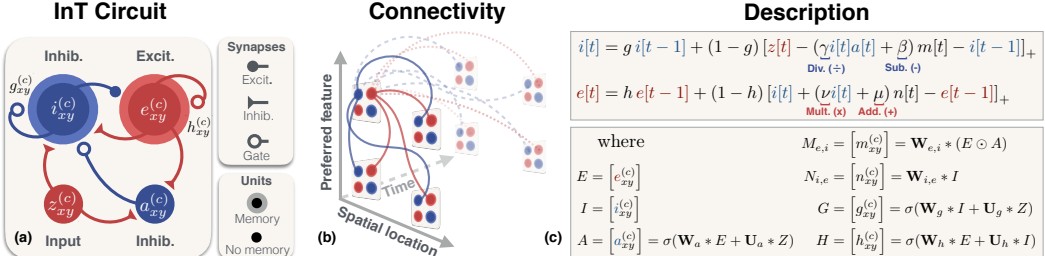

**Figure 4:** The Index-and-Track (InT) circuit model is inspired by Neuroscience models of motion perception [45] and executive cognitive functions [46]. (*a*) The circuit receives input encodings from a video ($z$), which are processed by interacting recurrent inhibitory and excitatory units ($i, e$) [7, 10], and a mechanism for selective "attention" ($a$) that tracks the target location. (*b*) InT units have spatiotemporal receptive fields. Spatial connections are formed by convolutions with weight kernels ($W_e, W_i$). Temporal connections are controlled by gates ($g, h$). (*c*) Model parameters are fit with gradient descent. Softplus= $[.]$, sigmoid= $\sigma$, convolution= $*$, elementwise product = $\odot$.

versions with the same number of frames, and the same or different number of distractors. In the first experiment, models were trained on the 32 frame and 14 distractor *PathTracker*, then tested on the 32 frame *PathTracker* datasets with 1, 14, or 25 distractors (Fig. 3a). In the second experiment, models were trained on the 64 frame and 14 distractor *PathTracker*, then tested on the 64 frame *PathTracker* datasets with 1, 14, or 25 distractors (Fig. 3a). Models were trained to detect if the target dot reached the blue goal marker using binary crossentropy and the Adam optimizer [44] until performance on a test set of 20K videos with 14 distractors decreased for 200 straight epochs. In each experiment, we selected model weights that performed best on the 14 distractor dataset. Models were retrained three times on learning rates $\in \{$1e-2, 1e-3, 1e-4, 3e-4, 1e-5$\}$ to optimize performance. The best performing model was then tested on the remaining 1 and 25 distractor datasets in the experiment. We used four NVIDIA GTX GPUs and a batch size 180 for training.

**Results** We treat human performance as the benchmark for models on *PathTracker*. Nearly all CNNs and the ImageNet-initialized TimeSformer performed well enough to reach the 95% human confidence interval on the 32 frame and 14 distractor *PathTracker*. However, all neural network models performed worse when systematically generalizing to *PathTracker* datasets with a different number of distractors, even when that number decreased (Fig. 3a, 1 distractor). Specifically, model performance on the 32 frame *PathTracker* datasets was worst when the videos contained 25 distractors: no CNN or transformer reached the 95% confidence interval of humans on this version of the dataset (Fig. 3a).

The optic flow R3D and the TimeSformer trained from scratch were less successful than the standard CNNs but still above chance, while the Conv-GRU performed at chance. The SORT Kalman filter was extremely sensitive to distractors, performing better than any other model on 1-distractor *PathTracker* datasets, but dropping well-below the human confidence interval on 14- and 25-distractor *PathTracker* datasets.

The performance of all models plummeted on 64 frame *PathTracker* datasets. The drop in model performance from 32 to 64 frames reflects a combination of the following features of *PathTracker*. (*i*) The target becomes more likely to cross a distractor when length and the number of distractors increase (Fig. 2; SI Fig. 2c). This makes the task difficult because the target is momentarily impossible to distinguish from a distractor. (*ii*) The target object must be tracked from start-to-end to solve the task, which can incur a memory cost that is monotonic w.r.t. video length. (*iii*) The prior two features interact to non-linearly increase task difficulty (SI Fig. 2c).

**Neural circuits for tracking without re-recognition** *PathTracker* is inspired by object tracking paradigms from Psychology, which tested theories of working memory and attention in human observers [2, 3]. *PathTracker* may draw upon similar mechanisms of visual cognition in humans. However, the video analysis models that we include in our benchmark (Fig. 3) do not have inductive biases for working memory, and while the TimeSformer uses a form of attention, it is insufficient for learning *PathTracker* and only reached human performance on one version of the challenge (Fig. 3).

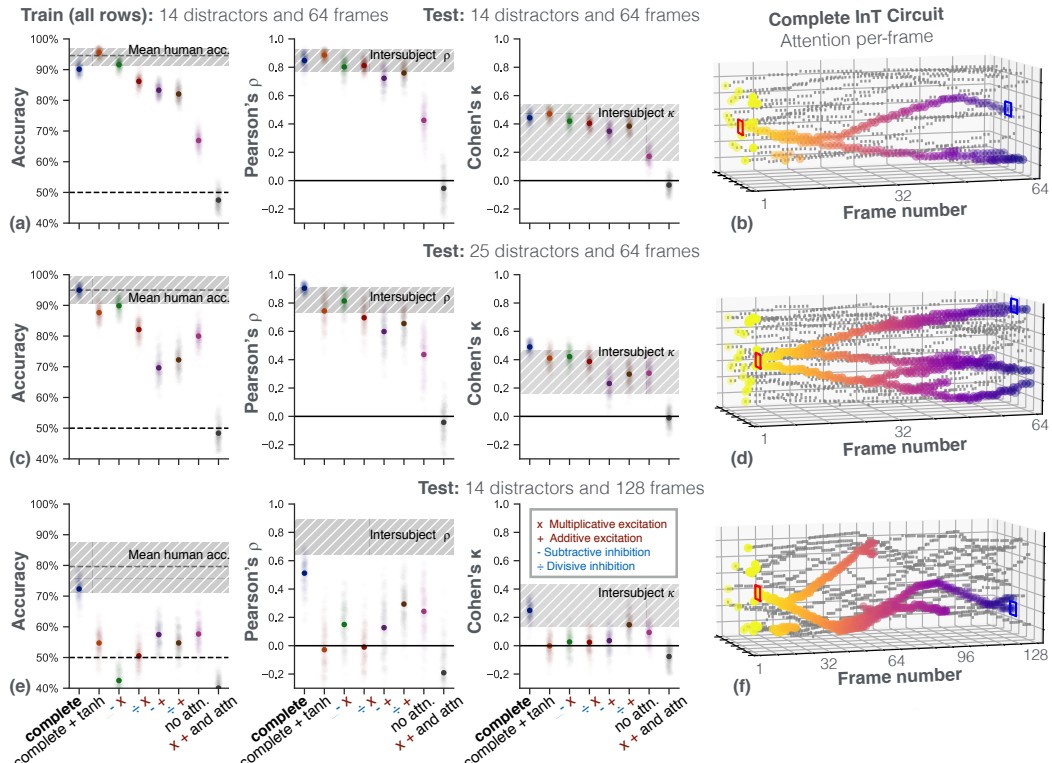

Figure 5: Performance, decision correlations, and error consistency between models and human observers on *PathTracker*. In a new set of psychophysics experiments, humans and models were trained on 64 frame *PathTracker* datasets with 14 distractors, and rendered decisions on a variety of challenging versions. Decision correlations are computed with Pearson's $\rho$, and error consistency with Cohen's $\kappa$ [59]. Only the Complete InT circuit rivals human performance and explains the majority of their decision and error variance on each test dataset (*a,c,e*). Visualizing InT attention (*a*) reveals that it has learned to solve *PathTracker* by multi-object tracking (*b,d,f*; color denotes time).

Neural circuits for motion perception, working memory, and attention have been the subject of intense study in Neuroscience for decades. Knowledge synthesized from several computational, electrophysiological and imaging studies point to canonical features and computations that are carried out by these circuits. (*i*) Spatiotemporal feature selectivity emerges from non-linear and time-delayed interactions between countervailing neuronal subpopulations [47–49]. (*ii*) Recurrently connected neuronal clusters can maintain task information in working memory [46,50]. (*iii*) Synaptic gating, inhibitory modulation, and disinhibitory circuits are neural substrates of working memory and attention [51–56]. (*iv*) Mechanisms for gain control may aid motion-based object tracking by building tolerance to visual nuisances, such as illumination [57,58]. We draw from these principles to construct the "Index-and-Track" circuit (InT, Fig. 4).

**InT circuit description** The InT circuit takes an input $z$ at location $x, y$ and feature channel $c$ from video frame $t \in T$ (Fig. 4a). This input is passed to an inhibitory unit $i$, which interacts with an excitatory unit $e$, both of which have persistent states that store memories with the help of gates $g, h$. The inhibitory unit is also gated by another inhibitory unit, $a$, which is a non-linear function of $e$, and can either decrease or increase (i.e., through disinhibition) the inhibitory drive. In principle, the sigmoidal nonlinearity of $a$ means that it can selectively attend, and hence, we refer to $a$ as "attention". Moreover, since $a$ is a function of $e$, which lags in time behind $z[t]$, its activity reflects the displacement (or motion) of an object in $z[t]$ versus the current memory of $e$. InT units have spatiotemporal receptive fields (Fig. 4b). Interactions between units at different locations are computed by convolution with weight kernels $\mathbf{W}_{e,i}, \mathbf{W}_{i,e} \in \mathbb{R}^{5,5,c,c}$ and attention is computed by $\mathbf{W}_a \in \mathbb{R}^{1,1,c,c}$. Gate activities that control InT dynamics and temporal receptive fields are similarly computed by kernels, $\mathbf{W}_g, \mathbf{W}_h, \mathbf{U}_g, \mathbf{U}_h \in \mathbb{R}^{1,1,c,c}$. Recurrent units in the InT support non-linear (gain) control. Inhibitory units can perform divisive and subtractive operations, controlled by $\gamma, \beta$. Excitatory units can perform multiplicative and additive operations, controlled by $\nu, \mu$. Parameters

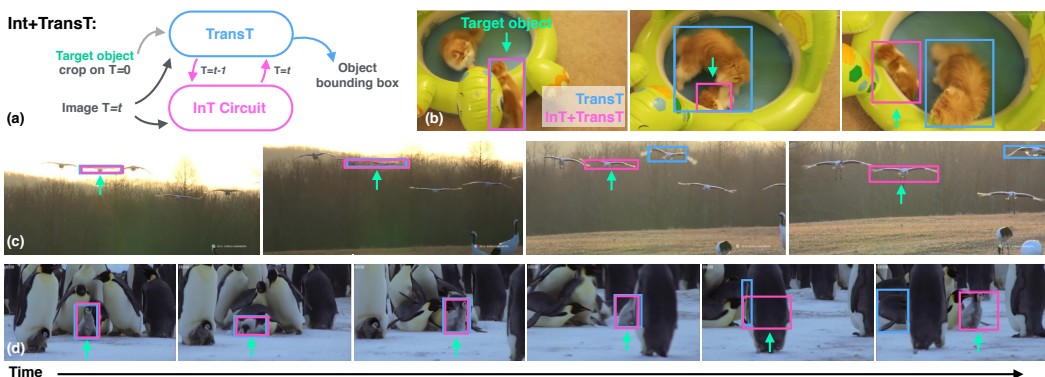

Figure 6: Circuit mechanisms for tracking without re-recognition build tolerance to visual nuisances that affect object appearance. (*a*) The TransT [60] is a transformer architecture for object tracking. We develop an extension, the InT+TransT, in which our InT circuit model recurrently modulates TransT activity. Unlike the TransT, the InT+TransT is trained on sequences to promote tracking strategies that do not rely on re-recognition. (*b-d*) The InT+TransT excels when the target object is visually similar to other depicted objects, undergoes changes in illumination, or is occluded.

$\gamma, \beta, \nu, \mu \in \mathbb{R}^c$. "SoftPlus" rectifications denoted by $[.]_+$ enforce inhibitory and excitatory function and competition (Fig. 4c). The final $e$ state is passed to a readout for *PathTracker* (SI §4).

**InT *PathTracker* performance** We trained the InT on *PathTracker* following the procedure in §4. It was the only model that rivaled humans on each version of *PathTracker* (Fig. 3). The gap in performance between InT and the field is greatest on the 64 frame version of the challenge.

How does the InT solve *PathTracker*? There are at least two strategies that it could choose from. One is to maintain a perfect track of the target throughout its trajectory, and extrapolate the momentum of its motion to resolve crossings with distractors. Another is to track all objects that cross the target and check if any of them reach the goal marker by the end of the video. To investigate the type of strategy learned by the InT for *PathTracker* and to compare this strategy to humans, we ran additional psychophysics with a new group of 90 participants using the same setup detailed in §3. Participants were trained on 8 videos from the 14 distractor and 64 frame *PathTracker* and tested on 72 videos from either the (*i*) 14 distractor and 64 frame dataset, (*ii*) 25 distractor and 64 frame dataset, or (*iii*) 14 distractor and 128 frame dataset. Unlike the psychophysics in §3, all participants viewing a given test set saw the same videos, which made it possible to compare their decision strategies with the InT.

InT performance reached the 95% confidence intervals of humans on each test dataset. The InT also produced errors that were extremely consistent with humans and explained nearly all variance in Pearson's $\rho$ and Cohen's $\kappa$ on each dataset (Fig. 5, middle and right columns). This result means that humans and InT rely on similar strategies for solving *PathTracker*.

What is the visual strategy learned by the InT? We visualized activity of $A$ units in the InT as they processed *PathTracker* videos and found they had learned a multi-object tracking strategy to solve the task (Fig. 5; see SI §5 for method, and http://bit.ly/InTcircuit for animations). The $A$ units track the target object until it crosses a distractor and ambiguity arises, at which point attention splits and it tracks both objects. This strategy indexes a limited number of objects at once, consistent with studies of object tracking in humans [2]. Since the InT is not explicitly constrained for this tracking strategy, we next investigated the minimal circuit for learning it and explaining human behavior.

We developed versions of the InT with lesions applied to different combinations of its divisive/subtractive and multiplicative/additive operations, a version without attention units $A$, and a version that does not make a distinction of inhibition vs. excitation ("complete + tanh"), in which rectifications were replaced with hyperbolic tangents that squash activities into $[-1, 1]$. While some of these models marginally outperformed the Complete InT on the 14 distractor and 64 frame dataset, their performance dropped precipitously on the 25 distractor and 64 frame dataset, and especially the very long 14 distractor and 128 frame dataset (Fig. 5e). Attention units in the complete InT's nearest rival (complete + tanh) were non-selective, potentially contributing to its struggles. InT performance also dropped when we forced it to attend to fewer objects (SI §5).

# 5  Appearance-free mechanisms for object tracking in the wild

The InT solves *PathTracker* by learning to track multiple objects at once, without relying on the re-recognition strategy that has been central to progress in video analysis challenges in computer vision. However, it is not clear if tracking without re-recognition is useful in the natural world. We test this question by turning to object tracking in natural videos. At the time of writing, the state-of-the-art object tracker is the TransT [60], a deep multihead transformer [61]. The TransT finds pixels in a video frame that match the appearance of an image crop depicting a target object. During training, the TransT receives a tuple of inputs, consisting of this target object image and a random additional "search frame" from the same video. These images are encoded with a modified ResNet50 [62], passed to separate transformers, and finally combined by a "cross-feature attention" (CFA) module, which compares the two encodings via a transformer key/query/value computation. The target frame is used for key and value operations, and the search frame is used for the query operation. Through its pure appearance-based approach to tracking, the TransT has achieved top performance on TrackingNet [13], LaSOT [42], and GOT-10K [43].

**InT+TransT**   We tested whether or not the InT circuit can improve TransT performance by learning a complementary object strategy that does not depend on appearance, or re-recognition. We reasoned that this strategy might help TransT tracking in cases where objects are difficult to discern by their appearance, such as when they are subject to changing lighting, color, or occlusion. We thus developed the InT+TransT, which involves the following modifications of the original TransT (Fig. 6a). (*i*) We introduce two InT circuits to form a bottom-up and top-down feedback loop with the TransT [10, 63], which in principle will help the model select the appropriate tracking strategy depending on the video: re-recognition or not. One InT receives ResNet50 search image encodings and modulates the TransT's CFA encoding of this search image. The other receives the output of the TransT and uses this information to update memory in the first InT. (*ii*) The TransT is trained with pairs of target and search video frames, separated in time by up to 100 frames. We introduce the intervening frames to the InT circuits. See SI §6 for extended methods.

**Training**   InT+TransT training and evaluation hews close to the TransT procedure. This includes training on the latest object tracking challenges in computer vision: TrackingNet [13], LaSOT [42], and GOT-10K [43]. All three challenges depict diverse classes of objects, moving in natural scenes that range from simplistic and barren to complex and cluttered. TrackingNet (30,132 train and 511 test videos) and GOT-10K (10,000 train and 180 test) evaluation is performed on official challenge servers, whereas LaSOT (1,120 train and 280 test) is evaluated with a Matlab toolbox. While the TransT is also trained with static images from Microsoft COCO [16], in which the search image is an augmented version of the target, we do not include COCO in InT+TransT since we expect object motion to be an essential feature for our model [60]. The InT+TransT is initialized with TransT weights and trained with AdamW [64] and a learning rate of $1e-4$ for InT parameters, and $1e-6$ for parameters in the TransT readout and CFA module. Other TransT parameters are frozen and not trained. The InT+TransT is trained with the same objective functions as the TransT for target object bounding box prediction in the search frame, and an additional objective function for bounding box prediction using InT circuit activity in intervening frames. The complete model was trained with batches of 24 videos on 8 NVIDIA GTX GPUs for 150 epochs (2 days). We selected the weights that performed best on GOT-10K validation. A hyperparameter controls the number of frames between the target and search that are introduced into the InT during training. We relied on coarse sampling (maximum of 8 frames) due to memory issues associated with recurrent network training on long sequences [11].

**Results**   An InT+TransT trained on sequences of 8 frames performed inference around 30FPS on a single NVIDIA GTX and beat the TransT on nearly all benchmarks. It is in first place on the TrackingNet leaderboard (`http://eval.tracking-net.org/`), better than the TransT on LaSOT, and rivals the TransT on the GOT-10K challenge (Table 5). The InT+TransT performed better when trained with longer sequences (compare $T = 8$ and $T = 1$, Table 5). Consistent with InT success on *PathTracker*, the InT+TransT was qualitatively better than the TransT on challenging videos where the target interacted with other similar looking objects (Fig. 6; `http://bit.ly/InTcircuit`).

We also found that the InT+TransT excelled in other challenging tracking conditions. The LaSOT challenge provides annotations for challenging video features, which reveal that the InT+TransT is

| Model | TrackingNet [13] | LaSOT [42] | GOT [43] | GOT Color | rColor | Occl. |
|---|---|---|---|---|---|---|
| InT+TransT$_{T=8}$ | **87.5** | 74.0 | 72.2 | 43.1 | 62.5 | 56.9 |
| InT+TransT$_{T=1}$ | 87.3 | 73.6 | 70.0 | 36.2 | 37.8 | 25.4 |
| TransT [60] | 86.7 | 73.8 | 72.3 | 40.7 | 57.5 | 55.2 |

Table 1: Model performance on TrackingNet ($P_{norm}$), LaSOT ($P_{norm}$), GOT-10K (AO), and perturbations applied to the GOT-10K (AO). Best performance is in black, and state of the art is bolded. Perturbations on the GOT-10K are color inversions on every frame ($Color$) or random frames ($rColor$), and random occluders created from scrambling image pixels ($Occl.$). InT+TransT$_{T=8}$ was trained on sequences of 8 frames, and InT+TransT$_{T=1}$ was trained on 1-frame sequences.

especially effective for tracking objects with "deformable" parts, such as moving wings or tails (SI §6). We further test if introducing object appearance perturbations to the GOT-10K might distinguish performance between the TransT and InT+TransT. We evaluate these models on the GOT-10K test set with one of three perturbations: inverting the color of all search frames (Color), inverting the color of random search frames (rColor), or introducing random occlusions (Occl.). The InT+TransT outperformed the TransT on each of these tests (Table 5).

## 6 Discussion

A key inspiration for our study is the centrality of visual motion and tracking across a broad phylogenetic range, via three premises: (*i*) Object motion integration over time *per se* is essential for ecological vision and survival [1]. (*ii*) Object motion perception cannot be completely reduced to recognizing similar appearance features at two different moments in time. In perceptual phenomena like *phi* motion, the object that is tracked is described as "formless" with no distinct appearance [65]. (*iii*) Motion integration over space and time is a basic operation of neural circuits in biological brains, which can be independent of appearance [66]. These three premises form the basis for our work.

We developed *PathTracker* to test whether state-of-the-art models for video analysis can solve a visual task when object appearance is ambiguous. Prior visual reasoning challenges like *Pathfinder* [7–9], indicate that this is a problem for object recognition models, which further serve as a backbone for many video analysis models. While no existing model was able to contend with humans on *PathTracker*, our InT circuit was. Through lesioning experiments, we discovered that the InT's ability to explain human behavior depends on its full array of inductive biases, helping it learn a visual strategy that indexes and tracks a limited number of the objects at once, echoing classic theories on the role of attention and working memory in object tracking [2, 3].

We further demonstrate that the capacity for video analysis without relying on re-recognition helps in natural scenes. Our InT+TransT model is more capable than the TransT at tracking objects when their appearance changes, and is the state of the art on the TrackingNet challenge. Together, our findings demonstrate that object appearance is a necessary element for video analysis, but it is not sufficient for modeling biological vision and rivaling human performance.

## Acknowledgments and Disclosure of Funding

We are grateful to Daniel Bear and Matthew Ricci for their suggestions to improve this work. We would also like to thank Rajan Girsa for help on Mechanical Turk experiments. GM is affiliated with Labrynthe Pvt. Ltd., New Delhi, India. Funding provided by ONR grant #N00014-19-1-2029, NSF grant (IIS-1912280), and the ANR-3IA Artificial and Natural Intelligence Toulouse Institute (ANR-19-PI3A-0004). Computing hardware supported by NIH grant S10OD025181 and the Center for Computation and Visualization (CCV) at Brown University.

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
