# Supplementary Information: Tracking Without Re-recognition in Humans and Machines

**Drew Linsley**[*1]**, Girik Malik**[*2]**, Junkyung Kim**[3]**,**

**Lakshmi N Govindarajan**[1]**, Ennio Mingolla**[†2]**, Thomas Serre**[†1]
drew_linsley@brown.edu and malik.gi@northeastern.edu

## 1 Limitations

In this work we tested a relatively small number of *PathTracker* versions. We mostly focused on small variations to the number of distractors and video length, but in future work we hope to incorporate other variations like speed and velocity manipulations, and generalization across temporal variations [1]. Another limitation is that appearance-free strategies confer relatively modest gains over the state of the art. One potential issue is determining when a visual system should rely on appearance-based vs. appearance-free features for tracking. Our solution is two-pronged and potentially insufficient. The first strategy is for top-down feedback from the TransT into the InT, which we aligns tracks between the two models. The second strategy is potentially naive, in that we gate the InT modulation to the TransT based on its agreement with the prior TransT query, and the confidence of the TransT query. Additional work is needed to identify better approaches. Meta-cognition work from Cognitive Neuroscience is one possible resource [2].

## 2 Extended Discussion

**Societal impacts**  The basic goal of our study is for understanding how biological brains work. *PathTracker* helps us screen models against humans on a simple visual task which tests visual strategies for tracking without "re-recognition", or appearance cues. The fact that we developed a circuit that explains human performance is primarily important because it makes predictions about the types of neural circuit mechanisms that we might ultimately find in the brain in future Neuroscience work. Our extension to natural videos achieves new state-of-the-art because it is able to implement visual strategies that build tolerance to visual nuisances in way that resembles humans. It must be recognized the further development of this model has potential for misuse. One possible nefarious application is for surveillance. On the other hand, such a technology could be essential for ecology, sports, self-driving cars, robotics, and other real-world applications of machine vision. We open source our code and data to promote research towards such beneficial applications.

## 3 Human benchmark

For our benchmark experiments we recruited 180 participants. Every participant was compensated with $8 through MTurk on successful completion of all test trials by pasting a unique code generated by the system into their MTurk account. The decision regarding this amount was reached upon by prorating the minimum wage. An additional overhead fee of 40% per participant was paid to MTurk. Collectively, we spent $1,440 on these benchmark experiments.

The experiment was not time bound and participants could complete it at their own pace, taking around 25 minutes to complete. Videos with 32-, 64- and 128-frames were of duration 4, 8 and 14 seconds respectively. The videos played at 10 frames per second. Participant reaction times were

35th Conference on Neural Information Processing Systems (NeurIPS 2021).

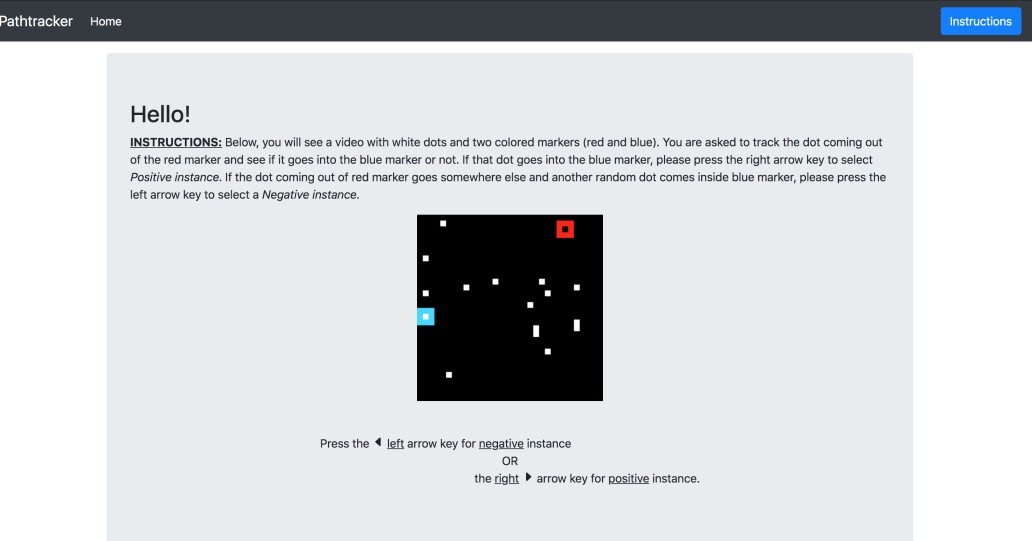

Figure 1: An experimental trial screen.

also recorded on every trial and we include these in our data release. After every trial participants were redirected to a screen confirming successful submission of their response. They could start the next trial by clicking the "Continue" button or by pressing spacebar. If not, they were automatically redirected to the next trial after 3000 ms. Participants were also shown a "rest screen" with a progress bar after every 10 trials where they could take additional and longer breaks if needed. The timer was turned off for the rest screen.

**Experiment design**    At the beginning of the experiment, we collected participant consent using a consent form approved by Brown University's Institutional Review Board (IRB). Our experiment was completed on a computer via Chrome browser. Once consented, we provided a demonstration clearly stating the instructions with an example video to the participants. We also provided them with an option to revisit the instructions, if needed, from the top right corner of the navigation bar at any point during the experiment.

Participants were asked to classify the video as "positive" (the dot leaving the red marker entered the blue marker) or "negative" (the dot leaving the red marker did not enter the blue marker) using the right and left arrow keys respectively. The choice for keys and their corresponding instances were mentioned below the video on every screen, along with a small instruction paragraph above the video. See Fig 1. Participants were given feedback on their response (correct/incorrect) after every practice trial, but not after the test trials.

**Setup**    The experiment was written in Python Flask, including the server side script and logic. The frontend templates were written in HTML with Bootstrap CSS framework. We used javascript for form submission with keys and redirections, done on the end-user side. The server was run with nginx on 1 Intel(R) Xeon(R) CPU E5-2695 v3 at 2.30GHz, 4GB RAM, Red Hat Enterprise Linux Server.

Video frames for each experiment were generated at $32 \times 32$ resolution. Before writing them to the mp4 videos displayed to human participants in the experiment, the frames were resized through nearest-neighbor interpolation to $256 \times 256$. In order to allow time for users to prepare for each trial, the first frame of each video was repeated 10 times before the rest of the video played.

**Filtering criteria**    Amazon Mechanical Turk data is notoriously noisy. Because of this, we adopted a simple and bias-free approach to filter participants who were inattentive or did not understand the task (these users were still paid for their time). For the main benchmark described in §3 in the main text, participants completed one of two experiments, where they were trained and tested on videos with 32 or 64 frames. No participant viewed both lengths of *PathTracker*. Participants were trained with 14 distractor videos, then tested on videos with 1, 14, or 25 distractors. We filtered

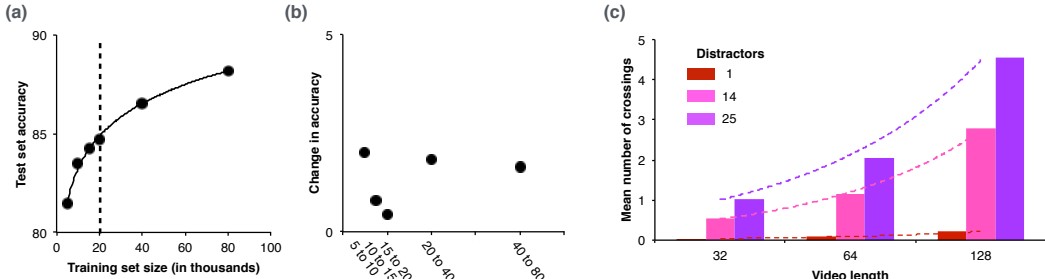

Figure 2: Our approach for selecting training set size on *PathTracker*, and a proxy for difficulty across the versions of the challenge. (*a*) We plot I3D performance as a function of training set size. The dotted line denotes the point at which the derivative of accuracy w.r.t. training set size is smallest (*b*). We take this change performance as a function of training set size as evidence that I3D has learned a strategy that is sufficient for the task. We suspected this size would make the *PathTracker* challenging but still solvable for the models we discuss in the main text. (*c*) The number of average crossings in *PathTracker* videos as a function of distractors and video length. Lines depict exponential fits for each number of distractors across lengths.

participants according to their performance on the *training videos* for a particular experiment, which were otherwise not used for any analysis in this study. We removed participants who did not exceed 2 median absolute deviations below the median $median(X) - 2 * MAD(X)$ (MAD = median absolute deviation [3]; this is a robust alternative to using the mean and standard deviation to find outliers). The threshold was approximately $40\%$ *training* accuracy for each experiment (chance is $50\%$). This procedure filtered 74/180 participants in the benchmark.

**Statistical testing**   We assessed the difference between human performance and chance using randomization tests [4]. We computed human accuracy on each test dataset, then over 10,000 steps, we shuffled video labels, and then recomputed and stored the resulting accuracy. We computed $p-$values as the proportion of shuffled accuracies that exceed the real accuracy. We also used linear models for significance testing of trends in human accuracy as we increased the number of distractors. From these models we computed $t$-tests and $p$-values.

**Using an I3D [5] to select *PathTracker* training set sizes**   As mentioned in the main text, we selected *PathTracker* training set size for models reported in the main text by investigating sample efficiency of the standard but not state-of-the-art I3D [5]. We were specifically interested in identifying a "pareto principle" in learning dynamics where additional training samples began to yield smaller gains in accuracy, potentially signifying a point at which I3D had learned a viable strategy (SI Fig. 2). At this point, we suspected that the task would remain challenging – but still solvable – across the variety of *PathTracker* conditions we discuss in the main text. We focus on basic 32 frame and 14 distractor training and find an inflection point at 20K examples. We plot I3D performance on this condition in SI Fig. 2a and performance slopes in SI Fig. 2b. The first and lowest slope corresponds to 20K samples, and hence may reflect an inflection in the model's visual strategy. Our experiments in the main text demonstrate that this strategy is a viable one for calibrating the difficulty of synthetic challenges.

**Target-distractor crossings**   We compute the number of average crossings between the target object and distractors in *PathTracker*. Increasing video length monotonically increases the number of crossings. Length further interacts with the number of distractors to yield more crossings (SI Fig. 2c).

## 4   Solving the Pathtracker challenge

**State-of-the-art model details**   We trained a variety of models on our benchmark. This included an R3D without any strides or downsampling. Because this manipulation caused an explosion in memory usage, we reduced the number of features per-residual block of this "No Stride R3D" from 64/128/256/512 to 32/32/32/32. We also included two forms of TimeSformers [6], one with distinct applications of temporal and spatial attention that we include in our main analyses, and another with join temporal and spatial attention (SI Fig. 3).

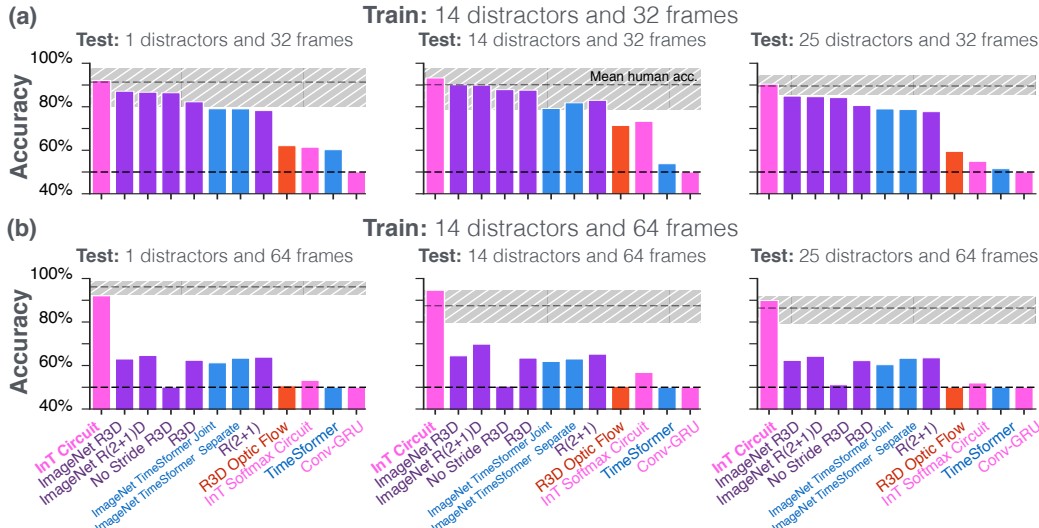

Figure 3: An extended benchmark of state-of-the-art models on *PathTracker* with (*a*) 32 and (*b*) 64 frame versions of the task.

**Optic Flow**    We followed the method of [5] to compute optic flow encodings of *PathTracker* datasets. We used OpenCV's implementation of the TV-L1 algorithm [7]. We extracted two channels from the output given by the algorithm, and appended a channel-averaged version of the corresponding *PathTracker* image, similar to the approach of [5].

# 5    InT circuit description

Our InT circuit has two recurrent neural populations, $I$ and $E$. These populations evolve over time and receive a dynamic "feedforward" drive via $Z$. This feedforward drive is derived from a convolution between each frame of the *PathTracker* videos a kernel $W_z \in \mathbb{R}^{1,1,3,32}$. This activity is then rectified by a softplus pointwise nonlinearity. InT hidden states are initialized with $0.6931 = \text{softplus}(0)$. The InT circuit also includes Batch Normalization [8] applied to the outputs of its recurrent kernels $W_i e, W_e i$, with scales ($\alpha$) and intercepts ($\eta$) shared across timesteps of processing. We initialize the scale parameters to $0.1$ following prior work [9]. We do not store Batch Normalization moments during training. InT gain control (i.e., its divisive normalization) is expected to emerge at steady state [10, 11] in similar dynamical systems formulations, although our formulation relaxes some of these constraints.

The final activity of $E[T]$ in the InT for a *PathTracker* video is passed to a readout that renders a binary decisions for the task. This readout begins by convolving $E[T]$ with a kernel $W_{r1} \in \mathbb{R}^{1,1,32,1}$. The output is channel-wise concatenated with the channel of the first frame containing the location of the goal marker. This activity is then convolved with another kernel $W_{r2} \in \mathbb{R}^{5,5,2,1}$, which is designed to capture overlap between the goal marker and the putative target object/dot. The resulting activity is "global" average pooled and entered into binary crossentropy for model optimization. On *PathTracker*, all versions of the InT and the ConvGRU used this input transformation. All versions of the InT, the ConvGRU, and the "No Stride R3D' used this readout.

**Spatiotemporal filtering through recurrent connections**    An open question is whether recurrent neural networks with convolutional connections are capable of learning tuned spatiotemporal feature selectivity. That is, the ability to learn to detect a specific visual feature moving in a certain direction. Adelson and Bergen [12] laid out a plausible solution, in which spatial filters offset by phase are combined over time through positive or negative weights. The success of our InT on *PathTracker* indicates that it might have adopted a similar solution, using its horizontal connection kernels $W_{i,e}, W_{e,i}$ to learn spatial filters offset in phase (e.g., an on-center off-surround and an off-center on-surround filter), which are combined via learned gates to yield spatiotemporal tuning. We leave an analysis of the InT "connectome" as it relates to spatiotemporal feature learning to future work.

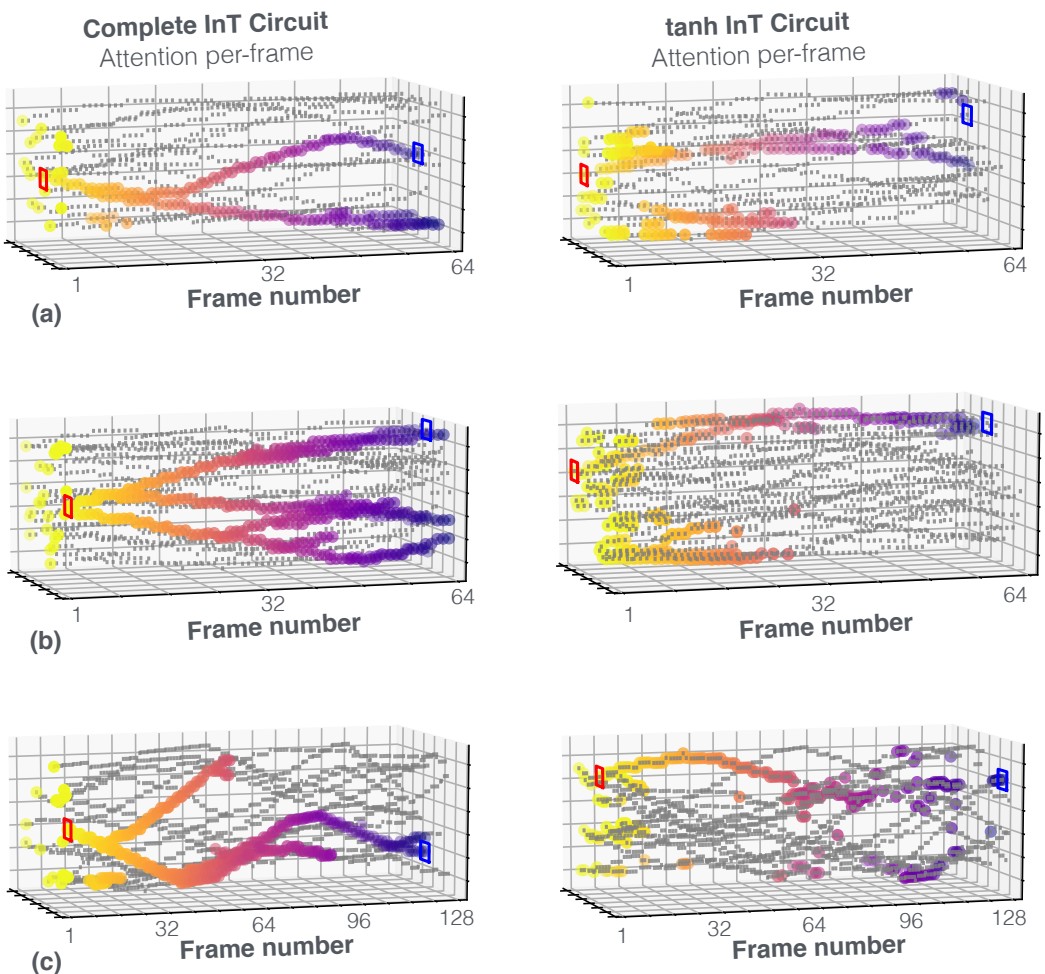

Figure 4: A comparison of attention between the complete InT and one where its softplus rectifications are replaced by tanh.

**Deriving the InT**  For the sake of clarity and succinctness, we focus the derivation of the $InT$ circuit's update equations to reflect that of generic single neurons, which without loss of generality applies to the each spatial/feature dimension. The $InT$ circuit model is built on top of two recurrent populations $(E/I)$ of neurons (serving excitatory/ inhibitory roles respectively), and a state-less population of neurons $(A)$ that serves as an attentional controller. We denote these populations as follows:

$$E = \left[e_{xy}^{(c)}\right]; I = \left[i_{xy}^{(c)}\right]; A = \left[a_{xy}^{(c)}\right] \tag{1}$$

Here, the $x, y$ subscripts denote spatial tuning, and the $c$ superscript denotes feature tuning. Moving forward, we reference generic units from these populations with $e$, $i$, and $a$ respectively. In essence, the circuit can be expressed as a continuous first-order coupled differential system of this form.

$$\tau_{inh}\frac{di}{dt} = -i + [z - (\gamma ia + \beta)m]_+$$
$$\tau_{exc}\frac{de}{dt} = -e + [i + (\nu i + \mu)n]_+ \tag{2}$$

In Eq. 2, $\gamma$, $\beta$, $\nu$, and $\mu$ are model hyperparameters, while $m$ and $n$ are themselves functions of $e$, $i$, and $a$. The exact functional form of $m$ and $n$ is detailed in Fig. 4b in the main text. $z$ is an external input to the system.

For the purposes of simulating and training this model with gradient descent, we use a first-order Euler approximation with time step $\Delta t$. Assuming we choose $g = \frac{\Delta t}{\tau_{inb}}$ and $h = \frac{\Delta t}{\tau_{exc}}$, the discretized version of Eq. 2 can be written as follows.

$$
\begin{aligned}
i_t &= (1 - g)\, i_{t-1} + g\left[z_t - (\gamma i_t a_t + \beta)\, m_t\right]_+ \\
e_t &= (1 - h)\, e_{t-1} + h\left[i_t + (\nu i_t + \mu)\, n_t\right]_+
\end{aligned}
\tag{3}
$$

Tuning the time constants $\tau_{exc}$, and $\tau_{inh}$ and choosing an appropriate $\Delta t$ can often prove to be tedious and challenging. To alleviate this, we introduce a "learnable" integration step, where $g$ and $h$ are modeled as neural gates. These are computed as specified in Eq. 4. $\sigma(.)$ is the sigmoidal function, which squashes activities in the range $[0, 1]$. $\mathbf{W}_g$, $\mathbf{U}_g$, $\mathbf{W}_h$, and $\mathbf{U}_h$ are convolutional kernels of size $1 \times 1 \times 32 \times 32$.

$$
\begin{aligned}
G &= \left[g_{xy}^{(c)}\right] = \sigma(\mathbf{W}_g * I + \mathbf{U}_g * Z) \\
H &= \left[h_{xy}^{(c)}\right] = \sigma(\mathbf{W}_h * E + \mathbf{U}_h * I)
\end{aligned}
\tag{4}
$$

## 6   Analyzing InT on *PathTracker*

We visualize InT $A$ attention units on *PathTracker* by simply binarizing the logits, where values greater than $mean(A[t]) + stddev(A[t])$ are set to 1 and units below that threshold are set to 0. When applying the same strategy to versions of the InT other than the complete circuit, we found attention that was far more diffuse. For this lesioned InT circuits, adjusting this threshold to be more conservative, choosing two or three or even four standard deviations above the mean, never yielded attention that looked like the complete model. For instance, the closest competitor to the complete InT is one in which its Softplus rectifications are changed to hyperbolic tangents, which remove model constraints for separate and competing forms of Inhibition and Excitation. This model's attention was subsequently diffuse and it also performed worse in generalization than the complete circuit (SI Fig. 4).

We also developed a version of the InT with attention that was biased against multi-object tracking. In the normal formulation, InT attention $A$ is transformed with a sigmoid pointwise nonlinearity. This independently transforms every unit in $A$ to be in $[0, 1]$, giving them the capacity to attend to multiple objects at once. In our version biased against multi-object tracking we replaced the sigmoid with a spatial softmax, which normalized the sum of units in each channel of $A$ to 1. This model performed worse than the CNNs or TimeSformer on *Pathtracker* (SI Fig. 3)

## 7   Extended *PathTracker* Decision Analyses

While the best performing 3D CNNs and Transformers we tested did not come close to human performance on 64-frame *Pathtracker* datasets, they were on par with humans on 32-frame *Pathtracker* datasets. To understand how well model decisions aligned with humans on these 32-frame *Pathtracker* datasets, we ran psychophysics on a set of 30 participants using the setup detailed in §3 of the main text. Participants were trained on 8 videos from the 14 distractor and 32 frame *PathTracker* and tested on 72 videos from the (*i*) 14 distractor and 32 frame dataset, (*ii*) 1 distractor and 32 frame dataset, or (*iii*) 25 distractor and 32 frame dataset. Like with the psychophysics used for Fig. 5 in the main text, all participants viewing a given test set saw the same videos so that we could compare their decision strategies with models.

We computed decision correlations between participants and the best-performing models on the 32-frame PathTracker videos: the R3D, Timesformer, and complete InT circuit (SI Table 1). We also computed 95% confidence intervals of inter-rater reliability for human participants, indicating their agreement for *PathTracker* videos. We did this by taking random split-half groups of the participants, computing the average decision for each video, then constructing distributions of decision correlations between the groups. Next, we computed the correlation between each model's sigmoidal decision

output and the average decision of humans. Only the InT's decision correlation with humans fell within the human confidence interval for all three versions of the 32-frame *PathTracker* dataset tested here. The R3D decision correlation also fell within the human confidence interval on the 1-distractor 32-frame *PathTracker* dataset, but not the 14- or 25-distractor versions. These results indicate that state-of-the-art video analysis models like the R3D and TimeSformer adopt different decision strategies than humans even when they achieve performance rivaling humans. The failure of these models to solve the 64-frame *PathTracker* datasets may reflect their bias away from learning visual strategies that are aligned with humans.

| Observer | 14 dist acc | 14 dist $\rho$ | 1 dist acc | 1 dist $\rho$ | 25 dist acc | 25 dist $\rho$ |
|---|---|---|---|---|---|---|
| Human | 95.83% | 0.93 | 92.19% | 0.92 | 100.00% | 0.96 |
| InT | 79.17% | 0.76$^\dagger$ | 95.83% | 0.94$^\dagger$ | 95.83% | 0.91$^\dagger$ |
| R3D | 75% | 0.62 | 100.00% | 0.95$^\dagger$ | 87.50% | 0.72 |
| TimeSformer | 83.33% | 0.68 | 79.17% | 0.63 | 83.33% | 0.69 |

Table 1: Performance and decision correlations between humans and models on the 32-frame *PathTracker* datasets. Models and humans were trained on a 14-distractor ("dist") version of the dataset and tested on 14-, 1- and 25-distractor videos. Model pearson correlations ($\rho$) falling within the 95%-bootstrapped confidence interval of human-to-human correlations are denoted by $\dagger$. Only the InT falls within the human confidence interval on all versions of the dataset.

## 8 InT+TransT

We modify a state-of-the-art tracker, TransT, with our InT circuit, to promote alternative visual strategies for object tracking (Fig. 5). We note that our InT+TransT model beats almost every benchmark metric on the LaSOT, TrackingNet, and GOT-10K object tracking challenges (SI Table 2).

| Method | Source | LaSOT | | | TrackingNet | | | GOT-10K | | |
|---|---|---|---|---|---|---|---|---|---|---|
| | | AUC | $P_{Norm}$ | P | AUC | $P_{Norm}$ | P | AO | $SR_{0.5}$ | $SR_{0.75}$ |
| InT+TransT | Ours | **65.0** | **74.0** | **69.3** | **81.94** | **87.48** | **80.94** | **72.2** | **82.2** | **68.2** |
| TransT | CVPR2021 | **64.9** | **73.8** | **69.0** | **81.4** | **86.7** | **80.3** | **72.3** | **82.4** | **68.2** |
| TransT-GOT | CVPR2021 | - | - | - | - | - | - | 67.1 | 76.8 | 60.9 |
| SiamR-CNN | CVPR2020 | 64.8 | 72.2 | - | 81.2 | 85.4 | 80.0 | 64.9 | 72.8 | 59.7 |
| Ocean | ECCV2020 | 56.0 | 65.1 | 56.6 | - | - | - | 61.1 | 72.1 | 47.3 |
| KYS | ECCV2020 | 55.4 | 63.3 | - | 74.0 | 80.0 | 68.8 | 63.6 | 75.1 | 51.5 |
| DCFST | ECCV2020 | - | - | - | 75.2 | 80.9 | 70.0 | 63.8 | 75.3 | 49.8 |
| SiamFC++ | AAAI2020 | 54.4 | 62.3 | 54.7 | 75.4 | 80.0 | 70.5 | 59.5 | 69.5 | 47.9 |
| PrDiMP | CVPR2020 | 59.8 | 68.8 | 60.8 | 75.8 | 81.6 | 70.4 | 63.4 | 73.8 | 54.3 |
| CGACD | CVPR2020 | 51.8 | 62.6 | - | 71.1 | 80.0 | 69.3 | - | - | - |
| SiamAttn | CVPR2020 | 56.0 | 64.8 | - | 75.2 | 81.7 | - | - | - | - |
| MAML | CVPR2020 | 52.3 | - | - | 75.7 | 82.2 | 72.5 | - | - | - |
| D3S | CVPR2020 | - | - | - | 72.8 | 76.8 | 66.4 | 59.7 | 67.6 | 46.2 |
| SiamCAR | CVPR2020 | 50.7 | 60.0 | 51.0 | - | - | - | 56.9 | 67.0 | 41.5 |
| SiamBAN | CVPR2020 | 51.4 | 59.8 | 52.1 | - | - | - | - | - | - |
| DiMP | ICCV2019 | 56.9 | 65.0 | 56.7 | 74.0 | 80.1 | 68.7 | 61.1 | 71.7 | 49.2 |
| SiamPRN++ | CVPR2019 | 49.6 | 56.9 | 49.1 | 73.3 | 80.0 | 69.4 | 51.7 | 61.6 | 32.5 |
| ATOM | CVPR2019 | 51.5 | 57.6 | 50.5 | 70.3 | 77.1 | 64.8 | 55.6 | 63.4 | 40.2 |
| ECO | ICCV2017 | 32.4 | 33.8 | 30.1 | 55.4 | 61.8 | 49.2 | 31.6 | 30.9 | 11.1 |
| MDNet | CVPR2016 | 39.7 | 46.0 | 37.3 | 60.6 | 70.5 | 56.5 | 29.9 | 30.3 | 9.9 |
| SiamFC | ECCVW2016 | 33.6 | 42.0 | 33.9 | 57.1 | 66.3 | 53.3 | 34.8 | 35.3 | 9.8 |

Table 2: Object tracking results on the LaSOT [13], TrackingNet [14], and GOT-10K [15] benchmarks. First place is in red and second place is in blue. Our InT+TransT model beats all others except for two benchmark GOT-10K scores.

**InT+TransT** We add two InT modules ($InT_1$ and $InT_2$) to the TransT architecture (Fig. 5). The key difference between these modules and the ones used on *PathTracker* is that they used LayerNorm [16] instead of Batch Normalization. This was done because object tracking in natural images is memory intensive and forces smaller batch sizes than what we used for *PathTracker*, which can lead to poor results with Batch Normalization.

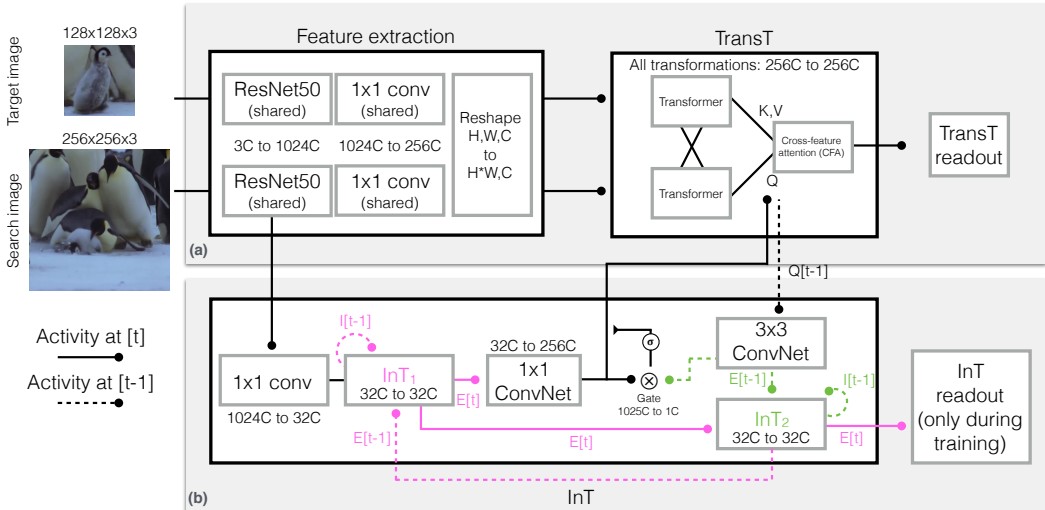

Figure 5: The (*a*) TransT and (*b*) InT addition to create our the InT+TransT. The InT additively modulate the TransT query (Q) in its CFA, which corresponds to its encoding of the search image which is compared to its encoding of the target. The InT activity is recurrent, and itself modulated by a "gate" which captures the similarity of InT activity and the TransT query from the prior step, along with the TransT query entropy. This gate shunts InT activity unless the TransT is low-confidence and the InT and TransT render different predictions, at which point the InT can adjust TransT queries. The InT is further supervised on each step of a video to predict target object bounding boxes.

$InT_1$ (Fig. 5b) has the same dimensionality as the one described for *PathTracker* in the main text. ResNet50 features $y \in \mathbb{R}^{1024 \times 32 \times 32}$ are reduced to $z \in \mathbb{R}^{32 \times 32 \times 32}$ by virtue of convolution with a kernel $W_{in} \in \mathbb{R}^{1 \times 1 \times 1024 \times 32}$, i.e., $z = y * W_{in}$. As input, $InT_1$ received $z$. A binary mask $B \in \mathbb{R}^{1 \times 32 \times 32}$ that specified the location of the target object in the very first frame was used to initialize the recurrent excitatory/inhibitory units of $InT_1$. They took values $E_{t=0} = B * W_{E_1}$ and $I_{t=0} = B * W_{I_1}$ respectively, where kernels $W_{E_1}, W_{I_1} \in \mathbb{R}^{1 \times 1 \times 1 \times 32}$, and $E_t, I_t \in \mathbb{R}^{32 \times 32 \times 32}$. The subscript $t$ for the recurrent population activities represent an arbitrary time point w.r.t. steps of processing.

To coregister the representations of $InT_1$ and $TransT$, we treat the excitatory units, $E_t$, of $InT_1$ by a transformation $f_\phi$ parameterized by three-layer convolutional neural network consisting of $1 \times 1$ kernels. $f_\phi$ essentially inflates dimensionality, i.e., $f_\phi(E_t) \in \mathbb{R}^{256 \times 32 \times 32}$. The network $f_\phi$ had softplus activation functions applied to the output of the first and second layers, and used kernels of dimensions $1 \times 1 \times 32 \times 256$, $1 \times 1 \times 256 \times 256$ and $1 \times 1 \times 256 \times 256$ in the three layers respectively. For notational convenience, we refer to $f_\phi(E_t)$ as $X_t$ in this discussion subsequently.

The "search frame" query ($Q_t$) for the TransT cross-feature attention (CFA) component was computed as a function of $X_t$ and $Q_{t-1}$ as described here. $Q_{t-1}$ was first subject to a transformation $f_\psi$, parameterized as another convnet, this time to register the query representation to the latent activities of the $InT$ modules. $f_\psi(Q_{t-1}) \in \mathbb{R}^{32 \times 32 \times 32}$ is used to compute two quantities: (a) a measure of spatial certainty in $Q_{t-1}$, and (b) a measure of spatial agreement between $Q_{t-1}$ and $E_t$. For spatial certainty we compute the channel wise $L^2$ norm of $f_\psi(Q_{t-1})$, yielding tensor $H^{(1)} \in \mathbb{R}^{1 \times 32 \times 32}$. For the spatial agreement measure, we compute the feature-wise outer product $H^{(2)} = f_\psi(Q_{t-1}) \otimes E_t \in \mathbb{R}^{1024 \times 32 \times 32}$. The mix-gate $G_{mix}$ was then a convolution on $H = \left[H^{(1)} H^{(2)}\right]$, with a kernel $W_{mix} \in \mathbb{R}^{1 \times 1 \times 1025 \times 1}$, followed by a sigmoidal non-linearity. The final TransT query $Q_t$ was then constructed as the sum of the original $Q_t$ and $G_{mix} \odot X_t$. Functionally, this mix-gate helps the InT+TransT compose a hybrid of appearance-free and appearance-based tracker query based on the intrinsic uncertainty of a video frame at a given moment in time. See SI Fig. 5 for a schematic.

The final step in the InT+TransT pipeline is "top-down" feedback from the TransT back to $InT_1$. This was done to encourage the two modules to align their object tracks and correct mistakes that emerged in one or the other resource [9]. $f_\psi(Q_{t=0})$, computed as described above, was used for initializing the excitatory units of $InT_2$ ($InT_2$ Fig. 5)b). The inhibitory units of $InT_2$ was initialized with $f_\psi(Q_{t=0}) * W_{I_2}$, where $W_{I_2} \in \mathbb{R}^{1,1,32,32}$. $E_t$ from $InT_1$ served as the input drive to $InT_2$ at

every time step $t$. To complete the loop, the recurrent excitatory state of $InT_2$ served as feedback for $InT_1$. We evaluated our InT+TransT on TrackingNet (published under the Apache License 2.0), LaSOT (published under the Apache License 2.0), and GOT-10K (published under CC BY-NC-SA 4.0). See SI Table 2 for a full comparison between our InT+TransT and other state-of-the-art models.

**Object tracking training and evaluation** The InT+TransT is trained with the same procedure as the original TransT, except that its InTs are given the intervening frames between the target and search images, as described in the main text. Otherwise, we refer the reader to training details in the TransT paper [17]. Evaluation was identical to the TransT, including the use of temporal smoothing for postprocessing ("Online Tracking"). As was the case for TransT, this involved interpolating the TransT bounding box predictions with a $32 \times 32$ Hanning window that penalized predictions on the current step $t$ which greatly diverged from previous steps. See [17] for details.

**InT+TransT error analysis on LaSOT** To systematically investigate object tracking errors of the InT+TransT vs. the Transt, we turned to the LaSOT object tracking dataset, which has annotations for different types of object transformations. According to LaSOT AUC, the InT+TransT is more tolerant than the TransT to appearance-perturbing transformations like clutter (0.588 InT vs. 0.579 TransT), object deformation (0.680 InT vs. 0.670 TransT), and illumination variations (0.663 InT vs. 0.652 TransT). In contrast, the TransT is more tolerant than the InT+TransT to fast motion (0.510 TransT vs. 0.492 InT), cases in which objects move off-screen and out-of-view (0.582 TransT vs. 0.574 InT), and is also slightly more tolerant to partial occlusion (0.620 TransT vs. 0.616 InT). Fast motion is an outlier quality that is more successfully handled by the TransT's pure appearance-based strategy than the InT's motion-based strategy. Partial occlusion is also a case where the InT's working memory-based tracking (see Fig. 6 in the main text and http://bit.ly/intcircuit for gifs) can be beaten by the TransT's appearance-based correspondence finding. Overall, these results suggest that there's further room for improvement of the InT+TransT – both in increasing its overall performance gap with the TransT and resolving the areas in which it performs worse. One possible path forward is to simply increase the depth of the InT that we include in the InT+TransT. Another possibility for improvement is to increase the granularity of the dynamics that the InT+TransT operates on. Achieving either of these goals will require extremely large-GPU memory nodes, or advances to recurrent learning algorithms with better memory complexity than standard backpropagation through time.