# OpenReview forum: "Tracking Without Re-recognition in Humans and Machines"
_NeurIPS.cc/2021/Conference — NeurIPS 2021 Poster_

### Official Review · Reviewer_JexJ · 2021-07-13

**Rating:** 7
**Confidence:** 5

**Summary:**

The paper presents a synthetically designed challenge called Pathtracker, which is to predict if a given green square reaches the desired destination. The idea is to abstract out the appearance and focus on the ability of the video analysis method to track a particular target among other similarly-looking distractors. The challenge is posed as a classification task (which requires tracking abilities beneath to solve it).


The paper proposes a circuit model in a recurrent architecture to solve the task and demonstrates superior performance to some of the existing SOTA methods for video analysis. Adequate human comparisons are made throughout the paper, in terms of the employed strategy as well as the final performance.  The experiment shows that combining the proposed circuit model with TransT leads to marginal improvements on large-scale object tracking benchmarks.

**Ethics Review Area:**

["I don’t know"]

**Limitations And Societal Impact:**

- There are differences in the literature, on how they approach single target vs multi-target tracking. Most multi-target tracking papers use a combination of appearance, motion (Kalman filters), spatial configuration/location, etc. Some of the recent papers suggest, that appearance can be learned unsupervised and may not be the strongest contributing factor in multi-target tracking (Karthik et al. Simple Unsupervised Multi-Object Tracking. https://arxiv.org/abs/2006.02609). A discussion in the paper would be useful. In fact, for me that makes the results in the paper not fully aligned with the motivation.

- A baseline experiment of applying simple Kalman filter-based tracking would be interesting to see. That might actually do well and I  would be keen to look at the results. Tackling tracking as tracking, rather than classification. By saying this, I am not downplaying the contributions. I just think the paper would look more complete with such an experiment.

- Another interesting direction would be to evaluate the method on cell tracking datasets (medical imaging). Such experiments would be better aligned with the primary motivation of the work.

- An experiment on a multi-target tracking dataset, just using Int would be useful (might be difficult to configure in rebuttal period, it is more of a future suggestion)

- Some writing suggestions and typos:
Figure1 is not clear. Is it tracking the jellyfish or the object below it.
Line 50 - Pathfinder are also exhibit
Venues are missing in many references (e.g. [7] is Neurips 2018)




**Main Review:**

The writing is clear and the paper is easy to read/comprehend. The Pathtracker challenge design and the following experiments appear novel to me. I liked the human comparisons and the strategy discussion with supporting experiments. The experiments look correct and reproducible.

To me, the experiments on the larger datasets like LASOT, GOT, etc. are not convincing enough. When the complexity increases, it becomes more and more difficult to judge the reason for the performance increase. Yes on the face, it appears the circuit model might be aiding performance, but possibly there are simpler improvements, which could lead to similar performance gains. I would have preferred the paper to focus on experiments in a different domain, something like cell tracking (medical), where the ideas would have been much more useful.






**Time Spent Reviewing:**

10

---

> ### Author Response · Authors · 2021-08-10
> **Our response.**
>
> Thank you for your review. You raised three issues:
>
> 1. You wanted a broader discussion of multi-object tracking in computer vision, in particular of the Karthik et al. paper that you mentioned. We included a discussion of multi-object tracking in the SI — it was not in the main text due to space constraints. But we will integrate this along with a discussion of Karthik et al. into the main text in our revision. Thank you for bringing this paper to our attention. It provides important background for appearance-based tracking and the success of leading models like the TransT.
>
> 2. You mention that a Kalman filter baseline would be interesting. We agreed and ran this experiment. We applied a Kalman filter with a Hungarian matching algorithm to frame-by-frame object detections in PathTracker.
>
> - We found that the Kalman filter can be tuned to work on 1-distractor datasets where the objects have linear motion and cross orthogonally. We have uploaded a gif at https://gfycat.com/incompletelankyaoudad. However, when the objects are allowed to move nonlinearly as they normally do in *PathTracker* the same Kalman filter failed. We have uploaded a gif of this failure to https://gfycat.com/impureinsignificantleopardseal. Indeed, Kalman filters rely on hand-tuning of several parameters to work well, and we found that these parameters needed to be adjusted for every *PathTracker* video to adjust to different object motion paths. Other changes and heuristics are needed to make the Kalman filters robust to different numbers of distractors in 1-, 14-, or 25-distractor versions of the challenge. Furthermore, nonlinear object motion and large numbers of distractors interact to make the problem even more challenging for finding the right set of parameters/heuristics to make a Kalman filter work. Because the *PathTracker* challenge involves training on 14-distractor videos (with 32- or 64-frames) then testing on 14-, 25-, and 1-distractor videos, we conclude that Kalman filters are too brittle to solve the challenge, explain human behavior, and ultimately allow us to extend insights from the challenge to real-world object-tracking.
>
> - The InT’s strategy for solving PathTracker gives us an idea of the flexibility needed for the challenge. This model has learned to index and track **any** object that’s deemed a potential target, for some amount of time. While it’s likely there’s ways to engineer Kalman filters that work on some PathTracker videos, they are not flexible enough for the entire challenge. Having this simple baseline is a fantastic addition to our paper. Even though Kalman filters are no longer the state-of-the-art on object tracking in real-world datasets, they are the classic method for object tracking, and their failure provides valuable motivation for our representation learning approach to solving PathTracker. We have also released our PathTracker dataset and video generation code to the general public, in case the community has modified Kalman filters to test. Thank you for this suggestion.
>
> 3. You mentioned that there’s issues with the formatting of references. We will fix this in our revision. You also mentioned that Figure 1 is unclear. The first row of images depicts two birds in a mating dance. We will clarify in the caption that they are birds. The yellow arrows point to the object that should be tracked (the male black & turquoise bird). We apologize for the confusion, because this figure is critical to our message that object tracking without re-recognition is an edge case of perception that is nevertheless ethologically essential.
>
> Finally, you raise excellent points about future applications of our InT+TransT to other visual domains, like cell tracking in biomedical imaging and multi-object tracking in real-world videos. We released all of our InT+TransT code with our paper in order to support these research directions, which we believe would benefit from our findings.

---

> > ### Comment · Reviewer_JexJ · 2021-08-16
> > **Kalman filter continued**
> >
> > I am not convinced with the Kalman filter experiment. A couple of qualitative examples do not really make the case. I was expecting a better analysis. I have a strong feeling that Kalman filtering with Hungarian matching will work equally well or maybe better than neural networks on the defined task. I am sharing an informal example here: https://www.youtube.com/watch?v=HQW4wtLBddk&ab_channel=StudentDave  (look at the end of the video). We have to approach it as a multi-target tracking problem.
> >
> > The movement limits in pathtracker between frames are pretty reasonable (2 pixels and 20 degrees) and it should work (compared to the shared example for instance).

---

> > > ### Author Response · Authors · 2021-08-16
> > > **Kalman filters**
> > >
> > > Thank you for the feedback!
> > >
> > > We applied the Kalman filter from the video you linked to 64-frame PathTracker videos. We tested the filter on 40 total videos: twenty videos with 1 distractor and twenty videos with 14 distractors. We supplied the filter with ground-truth object detections. We then evaluated classification accuracy by checking whether or not the object that starts in the “Start marker” is tracked into the “End marker”. You can find all of the code for the experiments along with visualizations in our repo: [https://github.com/pathtracker-code/kalman](https://github.com/pathtracker-code/kalman).
> > >
> > > Unlike humans or our InT, this Kalman filter struggled to track the target object when it crossed distractors. The filter was excellent at detecting when the target object reached the goal on the 1-distractor dataset (100% acc, 10/10 positive examples and 10/10 negative examples) but performed poorly on the 14-distractor dataset (75% accuracy, 5/10 positive examples and 10/10 negative examples). The filter's accuracy on the latter dataset is below human and InT performance. To be sure, the model produced some successful tracks, even in videos with many distractors (for example, [here](https://github.com/pathtracker-code/kalman/blob/master/multi_ex/vid_tracks_0.pdf) and [here](https://github.com/pathtracker-code/kalman/blob/master/multi_ex/vid_tracks_14.pdf)). But all too often the filter was rendered ineffective by clutter. These results are similar to what we found with our own Kalman filter implementation. We will extend our analysis of this model to the entire PathTracker challenge and add the results to the SI in our revision.
> > >
> > > As we mentioned in our previous response, we are sure it’s possible to tune Kalman filters for individual PathTracker videos, but our work is about learning visual strategies that generalize across the entire PathTracker challenge, like humans can, and will ultimately inform algorithmic strategies for solving real-world tracking.
> > >
> > > We are extremely grateful for your suggestion and we believe that this additional Kalman filter baseline adds a lot to the paper. We are also excited to see whether or not the community can develop Kalman filters that can rival humans on the PathTracker challenge. Even though Kalman filters are no longer the state of the art for modeling object tracking they serve as a important baseline for demonstrating that the tracking tasks we investigate in this paper are far from trivial.
> > >
> > > We look forward to hearing your thoughts!

---

> > > > ### Comment · Reviewer_JexJ · 2021-08-19
> > > > **Final discussion**
> > > >
> > > > Thanks for the experiments. I have two more thoughts:
> > > >
> > > > 1. Can you please try SORT instead and show some results (it also uses Kalman). SORT is one of the first baselines we usually try in MOT benchmarks (https://github.com/abewley/sort)
> > > >
> > > > 2. On the experiments on tracking datasets, some efforts augment simpler motion models with SOTA trackers. For instance, Wang et al. use the Kalman filter on top of SiamRPN (the best tracker at that time) https://arxiv.org/pdf/2007.01120.pdf . I feel marginal improvements can be obtained using a simpler motion model (similar to using the circuit model). I know such experiments will not be easy to do now and I am not asking for it. However, I would really hesitate to use the proposed circuit model on real-world tracking datasets. Any thoughts?
> > > >
> > > > Again, I appreciate the discussion put forth by the authors. I just want to be clear before I put down my final rating.

---

> > > > > ### Author Response · Authors · 2021-08-19
> > > > > **Responding to your points**
> > > > >
> > > > > We really appreciate your feedback, time, and effort in this reviewing process. Your suggestions to incorporate algorithms and background from Multi-object Tracking have helped us broaden the scope of the paper and provide better context for our findings.
> > > > >
> > > > > ---
> > > > >
> > > > > We implemented the *Sort* Kalman filter that you suggested [https://github.com/pathtracker-code/kalman/blob/master/sort.py](https://github.com/pathtracker-code/kalman/blob/master/sort.py). We first applied *Sort* to the 40 *Pathtracker* videos we used to validate the other Matlab-based Kalman filter. These were all 64-frame videos, 20 with one distractor, and 20 with fourteen distractors. *Sort* performed similarly to the Matlab-based Kalman filter, solving 95% (9/10 pos and 10/10 neg) one-distractor videos and 60% (2/10 pos and 10/10 neg) fourteen-distractor videos.
> > > > >
> > > > > Fortunately, the *Sort* algorithm is significantly faster than the Matlab-based Kalman filter from before. This made it possible to quickly check performance on datasets we used in the paper. We turned to the three datasets we described in Figure 5, in which we compared the performance of Humans and our InT circuit model. These datasets featured *Pathtracker* videos with 64 frames and 14 distractors, 64 frames and 25 distractors, and 128 frames and 14 distractors.
> > > > >
> > > > > *Sort* performed worse on these datasets than Humans across the board. It also performed worse than the InT even though *Sort* received ground-truth object detections whereas the InT was trained from scratch. Interestingly, *Sort* was relatively more sensitive to clutter than video length. This contrasts with Humans and the InT, who struggled more on longer videos than on shorter videos with more clutter.
> > > > >
> > > > > | Observer  | 64 frame 14 distractor                 | 64 frame 25 distractor | 128 frame 14 distractor |
> > > > > |----------------------|:----------------------------------------:|:------------------------:|:-------------------------:|
> > > > > | Human (*Fig. 5*)                  | 95%                                      | 95%                      | 79%                       |
> > > > > | InT (*Fig. 5*)                    | 92%                                      | 95%                      | 73%                       |
> > > > > | Kalman Filter \(*Sort*\) | 64%                                      | 53%                      | 58%                       |
> > > > >
> > > > > Importantly, your suggestions have made it clear that we must include the *Sort* algorithm as another baseline in our challenge. While it does not change our conclusions, it is a standard algorithm in the field of MoT and it reveals the effectiveness of a purely momentum-based strategy on our challenge. Finally, as we mentioned in our earlier responses, we suspect that there are specific Kalman filter parameterizations that can solve specific versions of *PathTracker* — even difficult videos with 25-distractors — but which may not be capable of generalizing. In our revision we will perform parameter sweeps to try and identify these "Goldilocks"-Kalman filters, and quantify how much parameters must change from one version of challenge to the next.
> > > > >
> > > > > ---
> > > > > You raise a good point about applying a Kalman filter to real-world tracking datasets. First, let us offer some background on the state-of-the-art TransT model that we used as the starting point for our Int+TransT. The TransT achieved state-of-the-art performance by relying solely on object appearance for tracking. On the GOT-10K, it posted a 11% improvement over the SiamR-CNN from CVPR 2020, a 22% improvement over the SiamFC++ from AAAI 2020, and a 46% improvement over the Siam-RPN from CVPR 2019.
> > > > >
> > > > > Our InT+TransT is novel because we introduce a dynamic InT into the TransT's architecture, to support its purely apperance-based tracking representations when these become uninformative. The entire model is trained end-to-end so that it can adjudicate between InT and transformer representations at any point in time. Our InT+TransT **outperforms** the TransT. It is the state of the art on TrackingNet ([http://eval.tracking-net.org/featured-challenges/39/leaderboard/42](http://eval.tracking-net.org/featured-challenges/39/leaderboard/42) see team *Circuit*) better on LaSOT, and is more robust to appearance-based perturbations on GOT-10K. Kalman filtering is a complementary operations to either the TransT or InT+TransT architecture.
> > > > >
> > > > > We believe that Kalman filtering can be introduced into either the TransT and InT+TransT as a post-processing step. The original TransT used a straightforward Hanning window smoothing operation as post-processing for bounding box predictions. We adopted this smoothing in the InT+TransT to make a fair comparison with the TransT. But we suspect that swapping this smoothing with Kalman filtering might improve performance. We are excited to explore this possibility in future work and continue to push the state of the art in object tracking.

---

> > > > > > ### Comment · Reviewer_JexJ · 2021-08-23
> > > > > > **Final note**
> > > > > >
> > > > > > I think that low-level methods can be on par with the proposed method on Pathtracker. However, the authors have comprehensively responded to my main concern, and I agree that probably more work is needed to achieve the same. Overall, I feel the paper is acceptable with some agreed changes, like adding some experiments and discussion on Kalman filter (maybe SORT as one of the baselines), also improving the discussion on MOT. I have increased my rating to 7.
> > > > > >
> > > > > > On a side note, I positively appreciate Neurips decision to go with openreview this year. Transparent discussions like these are crucial in the decision process.

---

### Official Review · Reviewer_utTc · 2021-07-16

**Rating:** 8
**Confidence:** 4

**Summary:**

The authors note that state-of-the-art computer vision methods for object tracking in natural videos largely rely on _appearanced-based tracking_ -- that is, some kind of matching between the features of a target object and features in subsequent frames of a movie. In contrast, people (and likely non-human animals) need to and are able to track objects in cases where appearance is unhelpful; they probably do this by somehow integrating the motion of an attended object over time.

Motivated by this discrepancy, the authors devise a tracking task in which object appearance is useless and find that people nevertheless perform well. They test a variety of neural network architectures (CNNs, RNNs, Transformers), none of which performs at human level on harder versions of the task. On the other hand, a custom-designed RNN with gating, inhibitory interactions, and "attention" both performs at human level and even makes similar errors to people. Finally, the authors show that they can improve on the state-of-the-art for several large-scale object tracking challenges by introducing their custom RNN as additional modules in the current best neural network.

**Ethical Concerns:**

No, I don't have ethical concerns.

**Limitations And Societal Impact:**

I don't see any specific mention of limitations or societal impact, but I'm not sure the manuscript/topic really calls for it. Short-term object tracking has been studied in people for decades; I don't think society would change in some predictably negative way if we understood human-level object tracking better (unlike, e.g., developing algorithms with super-human ability to recognize faces.)

I think if the authors could address my concerns above, it would adequately cover the "limitations" of their work.

**Main Review:**

I like this paper and this approach, and I hope that comparing models to actual human behavior on challenging tasks becomes more common at venues like NeurIPS. Even if the authors had not included the part at the end on "object tracking in the wild," the fact that standard ANN architectures don't reach human performance or follow the same strategies for object tracking on a simple task is interesting and worth communicating: it means that those standard architectures are not good models of all human visual abilities (though I have some questions about this experiment below.) I take the authors' main claims to be

(1) People can do a tracking task where object appearance cues are useless

(2) This task, in its more challenging variants, strains a number of standard ANN architectures, whether pretrained or not;

(3) A custom RNN architecture (InT) can perform the tracking task as well as people and adopts a similar algorithmic strategy to do so;

(4) Adding this architectural module to state-of-the-art object trackers improves them, suggesting that the computations learned by InT modules are useful "in the wild," not just for the task where object appearance is not informative.

Of these claims, I think the authors provide strong evidence for (1), pretty good evidence for (2) (but I have some concerns), very strong evidence for (3), and weak evidence for (4). So I'll focus my comments on (2) and (4) below.

_Claim 2: Are standard ANN architectures really unable to do the tracking task?_  It looks like many of the non-InT models do as well as humans -- or could do so with slight modification (more channels, more training, etc.) -- on the 32-frame version of the task (Figure 3a.) However, they're nowhere close on the 64-frame version. It's not intuitively obvious why this would be the case: the authors say the task difficulty should increase linearly with the number of frames (and they show that the number of distractor-target crossings increases only a little faster than linearly in the Supplement). This fact makes me a little worried that there's some critical hyperparameter that, if sampled thoroughly, might allow the standard ANNs to do a lot better on the 64-frame version. I have no way of knowing this for sure, but it seems worth trying to understand **why** the best standard ANNs fail in the longer version of the task. Are they learning a completely different algorithmic strategy from what people use, which just breaks down for longer movies? Is their memory capacity limiting? For some of the model architectures it may be hard to figure out exactly what's going on in natural language terms, but for others (like the Transformer architectures with their visualizable attention maps) maybe the authors could get some insight.

Related to this: even on the variants of the task where the standard ANNs succeed, how do they compare to humans in response/error pattern (i.e. using the metrics in Figure 5?) It would at least be suggestive if it turned out that those models were reaching human performance without making human-like errors, since that could mean that they're biased toward learning a strategy that just doesn't scale to longer movies.

_Claim 3: What makes InT so good?_ The authors' results with InT are very compelling, showing that this network reaches human performance and makes human-like responses. Their ablation study (Figure 5) also suggests that most if not all of the custom circuit components in InT are needed to get these results. That's most important, but I'm frankly a little surprised that the authors were able to engineer a recurrent circuit to do this task "from principles" (described in the Supplement) -- designing such highly customized circuits hasn't shown widespread success on most large-scale computer vision tasks. This doesn't mean the authors are wrong, but it does suggest to me that there's some key algorithmic strategy that the InT circuit is well-biased to learn that the other architectures they test aren't. Is this because the other architectures are too large? Missing some components (e.g. local recurrence or sigmoid activations for local "attention" or spatial information in the inputs?) Require more training?

It might be useful to try some "gain of function" experiments in which individual InT components are _added_ to the standard ANN architectures to see if this helps on the task. I especially wonder about the Conv-GRU, since it was the only other recurrent circuit tested: is it just incapable of (or biased against) learning the same algorithm as InT? If so, what needs to change?

_Claim 4: Is the InT circuit useful for tracking on large-scale computer vision benchmarks?_ The TransT + InT model does achieve state-of-the-art on 2/3 tracking benchmarks (albeit by small amounts) and seems more robust to the perturbations in the GOT task.
But these improvements are small enough that I wonder whether they're really due to the authors' custom architecture or to the changes they made to train it. Those changes are a little hard to follow in the text. For instance, it seems like InT + TransT gets additional frames at training time, maybe additional supervision signals (on the output of the InT module), and maybe additional inputs (the spatial location of the target object in the initial frame) compared to TransT. I also wasn't clear on whether it had approximately the same number of parameters (this isn't the most critical thing IMO, but it would be nice if TransT had a "control augmentation" that could be trained instead of the InT modules.) It would be very helpful if the authors could clarify all this in the main text and explain in more detail why they're making a fair comparison.

The three examples shown in Figure 6 are in good concordance with the authors' claims about the advantages of InT over purely recognition-based tracking (i.e., a number of objects with similar appearance that change positions a lot over the course of the movie and have occlusions.) However, since overall performance is not much better, there must also be some cases where TransT + InT performs worse (or else the ones where it does better are extremely rare.) It would be interesting to see some examples of these cases, or if possible do an analysis just on videos that have full occlusion/multiple objects of the same category. I'm not sure if annotations for these exist in the benchmark datasets, so if not the authors could think about creating small testing datasets to isolate these factors explicitly (not for a revision here, but in the medium-term.)


**Time Spent Reviewing:**

3

---

> ### Author Response · Authors · 2021-08-10
> **Our response.**
>
> Thank you for your review. Your feedback inspired us to perform several experiments that clarify the mismatch between humans and video analysis models on PathTracker and also strengthen our claims on real-world datasets.
>
>
> **Claim 2: Are standard ANN architectures really unable to do the tracking task?**
>
> Your questions about Claim 2 center around why video analysis models which perform well on 32-frame *PathTrackers* fail on 64-frame *PathTrackers*. You also made the adroit suggestion of addressing this question by comparing decision strategies of video analysis models and humans on the 32-frame *PathTrackers*. During this rebuttal period we ran your suggested experiment on 30 Mturk participants. Each participant was trained to solve 32-frame 14-distractor videos then tested on 32-frame videos with 14-distractors, 1 distractor, and 25-distractors (24 videos per condition; see the psychophysics methods in the paper for details on the experimental setup). We then computed decision correlations between participants and the best-performing models on the 32-frame *PathTracker* videos: the R3D, Timesformer, and complete InT. Results are in the table below; significant differences are denoted by asterisks (**=p$<$0.01, ***=p$<$0.001).
>
> | **Observer** | **14-distractor Acc** | **14-distractor Pearson**       | **1-distractor Acc** | **1-distractor Pearson**    | **25-distractor Acc** | **25-distractor Pearson**         |
> |------------|:---------------------:|:-------------------------------:|:--------------------:|:---------------------------:|:---------------------:|:---------------------------------:|
> | Human        | 95\.83%               | 0\.93                           | 92\.19%              | 0\.92                       | 100\.00%              | 0\.96                             |
> | InT          | 79\.17%               | 0\.76\*\* \(R3D & Timesformer\) | 95\.83%              | 0\.94\*\*\* \(Timesformer\) | 95\.83%               | 0\.91\*\*\* \(R3D & Timesformer\) |
> | R3D          | 75%                   | 0\.62                           | 100\.00%             | 0\.95\*\*\* \(Timesformer\) | 87\.50%               | 0\.72                             |
> | Timesformer  | 83\.33%               | 0\.68                           | 79\.17%              | 0\.63                       | 83\.33%               | 0\.69                             |
>
> InT decisions are significantly more correlated with humans than the R3D or Timesformer on the 14- and 25-distractor datasets. InT and R3D decisions are both significantly more correlated with humans than the Timesformer on the 1-distractor dataset. As you hypothesized, SOTA video analysis models like the R3D and Timesformer adopt different decision strategies than humans even when they’re able to rival human performance. This finding suggests that the failure of these models to solve the 64-frame *PathTrackers* reflects their bias towards learning a visual strategy that is inefficient and mismatched with humans.
>
>
> **Claim 3: What makes InT so good?**
> You wanted to understand why the InT was so effective on *PathTracker*, and you suggested that “gain of function” experiments could identify the essential model components. Thank you for this suggestion, because we were not clear enough in our description of this result in the original manuscript.
>
> We describe “gain of function” for InT mechanisms on *PathTracker* in Figure 5. In the left column of this figure we plot accuracy of different configurations of the InT on difficult versions of *PathTracker*. The model called “x + and attn” is a version of the InT with attention and excitatory units capable of multiplicative (x) and additive (+) operations. This model is equivalent to a ConvGRU with additive and multiplicative computations for its hidden state update. It has an input gate, which we consider “attention”, that is a function of excitatory units and implemented with a 1$\times$1 convolutional layer. This ConvGRU-like model is at chance accuracy on the 14 distractor and 64 frame *PathTracker*. By removing this form of attention and introducing  separate populations of inhibitory/excitatory neurons, performance improves to ~65%. After introducing disinhibitory attention (see Figure 4) performance jumps to >=85%. In other words, Figure 5 reveals the two most important mechanisms needed for a ConvGRU to perform well are (i) countervailing populations of I/E recurrent neurons, and (ii) disinhibitory attention.
>
>
> **Claim 4: Is the InT circuit useful for tracking on large-scale computer vision benchmarks?**
>
> To strengthen our claim that the InT can help object tracking in real-world video you asked us to clarify model details, perform additional controls, and analyze the mistakes of the InT+TransT vs. the TransT.
>
> The InT+TransT and the original TransT get the same supervision signals. There are two key differences in training, neither of which gives the InT+TransT more ground truth information than the TransT. During training, the TransT receives a crop-box of the target object in a frame at time $t$ and a “search” frame at time $t+n$, where $n$ is a random positive or negative integer that is bounded by the length of a given video. The model is trained to localize the target object in the search frame. For the InT+TransT, $n$ is a strictly positive number, and the model is given intervening search frames between $t$ and $t+n$. The number of these search frames is denoted in Table 1 in the main text ($T=8$ or $T=1$). When the number of intervening search frames is greater than 1, the model is trained with backpropagation through time in order to propagate gradients from the loss computed only at $t+n$ through the InT across the sequence. As you point out, the hidden state in the InT is primed with the location of the target object since the model is designed for tracking without appearance cues.
>
> We trained new control models to better understand whether or not the InT is helpful for tracking objects in real-world videos. We first trained a version of the InT+TransT where we replaced the complete InT with the ConvGRU-equivalent version we describe above (“x + and attn” from Figure 5). This control model performed worse than the original InT+TransT on every tracking challenge we tested. In the original manuscript we also included another pertinent control, the  InT\+TransT_${T=1}$, which uses 1 timestep rather than the 8 in the complete InT+TransT. This 1-timestep model serves as the “control augmentation” model you requested: it is a purely feedforward model, where the InT provides the TransT additional capacity rather than an ability to track objects by their motion. We will include these controls in our revision.
>
> | **Model**    | **TrackingNet** | **LaSOT** | **GOT** | **GOT Color** | **GOT rColor** | **GOT Occl.** |
> |------------|:---------------:|:---------:|:-------:|:-------------:|:--------------:|:-------------:|
> | InT\+TransT_${T=8}$  | 87\.48          | 0\.74     | 0\.722  | 0\.431        | 0\.625         | 0\.569        |
> | InT\+TransT_${T=1}$  | 87\.29          | 0\.736    | 0\.7    | 0\.362        | 0\.378         | 0\.254        |
> | TransT       | 86\.69          | 0\.738    | 0\.723  | 0\.254        | 0\.575         | 0\.552        |
> | ConvGRU\+TransT | 86\.21          | 0\.714    | 0\.711  | 0\.395        | 0\.577         | 0\.508        |
>
> Finally, you asked about differences in errors between the InT+Transt and the original TransT. We focused our discussion in the manuscript on appearance-driven errors of the TransT which the InT+TransT corrects. To systematically investigate the opposite case, we turned to the LaSOT object tracking dataset, which has annotations for different types of object transformations. According to LaSOT AUC, the InT+TransT is more tolerant than the TransT to appearance-perturbing transformations like clutter (0.588 InT vs. 0.579 TransT), object deformation (0.680 InT vs. 0.670 TransT), and illumination variations (0.663 InT vs. 0.652 TransT). In contrast, the TransT is more tolerant than the InT+TransT to fast motion (0.510 TransT vs. 0.492 InT), cases in which objects move off-screen and out-of-view (0.582 TransT vs. 0.574 InT), and is also slightly more tolerant to partial occlusion (0.620 TransT vs. 0.616 InT). Fast motion is an outlier quality that is more successfully handled by a pure appearance-based strategy than the InT’s motion-based strategy. TransT has a postprocessing strategy for handling objects when they move off-screen that we did not re-tune for our model, but evidently needs to be balanced with the InT’s learned strategy for handling occluded objects.  Partial occlusion is also a case where the InT’s working memory-based tracking (see Fig. 6 in the main text and http://bit.ly/intcircuit for gifs) can be beaten by the TransT’s appearance-based correspondence finding. Overall, these results suggest that there’s further room for improvement in the InT+TransT! We will include these results in the SI of our revision along with ideas for future directions of the InT+TransT, such as deeper InTs trained on longer sequences.

---

> > ### Comment · Reviewer_utTc · 2021-08-30
> > **Thank you!**
> >
> > Thank you for your clear response (and additional experiments.) I agree that the results from the additional human experiments suggest the non-InT models are adopting non-human-like strategies, even when they're accurate. It's a little strange that the 14-distractor condition seems to be more challenging than the 1- and 25-distractor conditions, but this could be due to the small number of videos in the test. In any case, I think it supports your claim.
> >
> > The further controls on tracking-in-the-wild are also interesting and clarifying. It seems that the improvement from adding InT to TransT is small but real (thank you for pointing out the parameter-matched control.) The analysis of which trials/conditions give the most improvement, versus ones where the original TransT is better, help explain what's going on -- and I agree they suggest further room for improvement on some "corner cases" (like very fast motion) where I'm not even sure what the "human-like" behavior is.
> >
> > I have also read the other reviewers' comments and your responses -- I think adding the non-neural network controls also substantially strengthens your work.

---

### Official Review · Reviewer_hxQK · 2021-07-20

**Rating:** 3
**Confidence:** 5

**Summary:**

This paper presents a new tracking challenge specially for testing the motion-based target tracking capability. "Objects" in the synthetic data are with the same appearance so as to exclude the appearance information. Human study and baseline methods are proposed to show how challenging this task is.

**Limitations And Societal Impact:**

Please refer to the detailed review above.

------------------Post Rebuttal-------------------------

The response does not address my concerns. The experiments on Page 8-9 are conducted on existing conventional visual object tracking datasets, where the target object usually has a different appearance from backgrounds/distractors. However, this paper aims to address the tracking problem when appearance has little if not no difference. A convinced evaluation in real-world scenarios should be to track an instance with distractors of the same category, such as a fish/sheep swimming/running surrounded by similar-looking fish/sheep as distractors. The current experimental results on real-world tracking datasets (Table 1) show only marginal improvements (< 1% on two out of three datasets, and -0.1% on the remaining one), and please note that the improvements are achieved under the condition that the tracking target has large appearance difference from distractors.

Overall, the proposed InT only shows effectiveness on toy synthetic data (the dataset scale is far smaller than existing real-world tracking datasets), and there are too many factors that may affect the results on synthetic data, such as the motion pattern of the distractors, the variation in the number of distractors as the video goes. And crutial experiments are missing (as mentioned above but not limitted to). Thus, I render a rejection for this paper.

**Main Review:**

This paper aims to emphasize the importance of the motion information of tracking targets and construct a corresponding synthetic dataset. However, the crucial rule of motion has already been noticed and explored in the conventional object tracking tasks, such as [1,2,3], although appearance information is also an important clue in those tasks.

Actually, even two objects are visually very similar from we human views (eg., cells), sometimes deep neural networks can still distinguish them. I think the designed task goes a bit to extremes. To avoid this, using real-world tasks would be better, such as tracking cells or fish.

Overall, I do not think this task has much value.

[1] Deep Motion Features for Visual Tracking.
[2] Visual Object Tracking for Unmanned Aerial Vehicles: A Benchmark and New Motion Models
[3] Robust Object Tracking based on Temporal and Spatial Deep Networks

**Time Spent Reviewing:**

3

---

> ### Author Response · Authors · 2021-08-10
> **Our response.**
>
> Thank you for your review. It appears that you are not convinced *PathTracker* is a worthwhile challenge.
>
> >”This paper presents a new tracking challenge specially for testing the motion-based target tracking capability… Overall, I do not think this task has much value.”
>
> We ask you to consider that the *PathTracker* challenge dataset is just one element of our work. We summarize our approach and contributions:
>
> - Biological visual systems can track objects even when appearance is uninformative. This is not an edge case. There are many biological examples of the need to follow the “world line” of the motion of one among several identically-appearing objects. This ability has been studied for more than half a century, and there is no real debate on this being a principal cognitive strategy. What we do in this work is to test if high-performing machine vision systems are similarly capable.
> - We investigate this question with large-scale assays of SOTA video analysis models and humans on PathTracker. While humans easily solve all versions of the task, no computer vision model before ours is capable of doing so.
> - To address the failure of SOTA models we introduce the InT, a circuit-model with mechanisms for motion processing, working memory, and attention. The InT solves PathTracker and explains the majority of variance in human decision making.
> - We extend the InT to **real-world** object tracking datasets and establish a new state of the art. Thus, developing a model to explain object tracking without re-recognition in biological vision ultimately helped improve the performance of artificial vision.
>
> We will include a discussion of the three papers you mention in our revision.
> - [1] Introduced a model that fuses optic flow and appearance features for object tracking in real-world data.IOur paper demonstrates that optic flow features are insufficient for solving PathTracker. The model in this paper is not state of the art.
> - [2] This paper explored object tracking when the camera undergoes extreme self-motion. They used particle filtering to explicitly model camera and object motion. These classic hand-tuned motion models struggle on PathTracker, and are no longer the state of the art in computer vision.
> - [3] The paper introduced a module (“Temporal Net”) for updating the appearance of the target object template throughout a video. The model in this paper is no longer state of the art. These days, leading object tracking models like the TransT are trained to build invariance to the types of object transformations that the Temporal Net tries to address.
>
> We look forward to further discussion.

---

> ### Author Response · Authors · 2021-09-01
> **Response to your Post-Rebuttal comments**
>
> Dear Reviewer,
>
> Thank you for your feedback.
>
> - **Our model is the state-of-the-art in object tracking (see team Circuit at [http://eval.tracking-net.org/featured-challenges/39/leaderboard/42](http://eval.tracking-net.org/featured-challenges/39/leaderboard/42)).** Reviewer **utTc** asked us to measure what kinds of visual perturbations in real-world video our InT+TransT handles better than the prior state of the art (TransT). We wrote, "According to LaSOT AUC, the InT+TransT is more tolerant than the TransT to appearance-perturbing transformations like clutter (0.588 InT vs. 0.579 TransT), object deformation (0.680 InT vs. 0.670 TransT), and illumination variations (0.663 InT vs. 0.652 TransT)." These improvements are sensible for a model that performs well on *PathTracker*. Reviewer **utTc** also asked us for additional control experiments to demonstrate that InT's performance is not due to extraneous factors like number-of-parameters or other idiosyncrasies introduced into training. Please see our response to that reviewer's "Claim 4", where we show that the full InT circuit is important for reaching state-of-the-art performance.
>
>
> - We tested our model on the same popular and large-scale object tracking datasets as the prior state of the art, the TransT. Target objects in these benchmarks can look quite similar to distractors. For instance, one might need to track one of two cats that are playing together [https://gfycat.com/poortepidclam](https://gfycat.com/poortepidclam). Or to track a target object (baby grey penguin) that is briefly occluded by clutter (adult penguin in the foreground) [https://gfycat.com/nearteeminggreendarnerdragonfly](https://gfycat.com/nearteeminggreendarnerdragonfly). These examples are quite similar to what you suggested. In these gifs, our InT+TransT tracks are in pink, whereas the prior state of the art is in Blue.
>
>
> - We'd also like to mention that reviewer **JexJ** pushed us to include additional baselines on *PathTracker* to make the task more relevant to the field of multi-object tracking. In our response to that reviewer we showed how even MoT baselines, like SORT, struggle to solve the task as well as humans *or* our InT circuit.
>
>
> - You write "the proposed InT only shows effectiveness on toy synthetic data (the dataset scale is far smaller than existing real-world tracking datasets)." This is incorrect. For instance, the GOT-10K has 10,000 videos, whereas datasets in our *PathTracker* challenge have 20,000 videos for training and 20,000 videos for testing (see line 161 in our paper).
>
>
> - You also write "and there are too many factors that may affect the results on synthetic data, such as the motion pattern of the distractors, the variation in the number of distractors as the video goes." As *PathTracker* is a synthetic challenge we can precisely control the factors that contribute to its difficulty. This helps us understand how the amount of clutter or video length affects the learning and inference of different observers and models — from humans to 3D-CNNs and from Transformers to Kalman filters. This precise control over data is essential for rigorously understanding how vision works.
>
>
> - Finally, you write "And crutial experiments are missing (as mentioned above but not limited to)." We hope you will agree from the gifs above, our responses to the other reviewers, and the extensive experiments in the paper, that our model outperforms the prior state-of-the-art object tracker on natural videos. We are happy to consider any other experiments you might have in mind, and look forward to your clarification.

---

> > ### Comment · Reviewer_hxQK · 2021-09-02
> > **Final discussion**
> >
> > Thanks for the response.
> >
> > I am not convinced by the results on the real-world tracking datasets. The improvements (0.8% on TrackingNet, 0.2% on LaSOT, and -0.1% on GOT) are rather marginal. They just seem like a minor fluctuation when adding something into a baseline. In addition, these datasets should be more easier for the proposed motion module, because they have sparser objects and larger appearance differences of the target from distractors. One can have a look at [here](https://medium.com/deep-learning-digest/lasot-large-scale-dataset-for-object-tracking-models-a4ff0eb03611) for LaSOT, [here](http://got-10k.aitestunion.com/index) for GOT, and [here](https://tracking-net.org/) for TrackingNet. If the proposed motion module really works, significant improvements on these datasets are expected regardless that these data are not suitable to test the proposed no-appearance-interfered motion module.
> >
> > Regarding the dataset scale, most of TrackingNet/LaSOT/GOT videos have tens of hundreds of frames, while the synthetic dataset is with 32/64/128 frames. Thus, the actual data in the existing tracking dataset for training or testing are much bigger, let alone TrackingNet has 30K videos.
> >
> > To summarize, based on the results on real-world object tracking datasets, I reckon that the motion module is not really effective. This might be caused that the synthetic dataset cannot reflect the real challenge in real-world object tracking scenarios. As I suggested in the previous review, the motion module is also not evaluated on suitable datasets. Thus, I keep the rejection decision.

---

> > > ### Author Response · Authors · 2021-09-02
> > > **Our response**
> > >
> > > If we were randomly tweaking the prior state of the art we would agree that a percentage point of improvement might not constitute an intellectual contribution. However, our InT circuit is first and foremost designed to solve the synthetic PathTracker challenge and explain human decision making when tracking without re-recognition.
> > >
> > > Our primary contribution is that we find that humans are far more capable of tracking objects without re-recognition than state-of-the-art models for video analysis. Our neurally-grounded InT circuit bridges this gap, solving *PathTracker* like humans and generating testable Neuroscience hypotheses.
> > >
> > > **The performance improvements we find on large-scale real-world object tracking datasets are concomitant with the InT's ability to explain human perception.** This is our secondary contribution and means that there is ecological utility for mechanisms humans rely on to track objects without re-recognition. Please see Section 5 of our paper and our discussion with reviewer **utTc** for more details on how tracking without re-recognition is a feature rather than a bug of human perception.
> > >
> > > As the reviewer points out, real world tracking datasets are not the best testbed for tracking without re-recognition because the target and background often contrast (although not always, as we show in the gifs in our previous post). This criticism is precisely what motivated us to develop *PathTracker*, and is really a criticism of standard real world tracking datasets rather than our work. Note that our appearance-perturbation experiments on the GOT-10K (Table 1) demonstrate that our InT+TransT is capable of huge improvements over the prior state of the art when visual nuisances make pure appearance-based tracking less effective.
> > >
> > > To address your comment regarding the scale of *PathTracker*: we selected the dataset size according to the results of pilot experiments we describe in Section 4. The link you provided for LaSOT ([here](https://medium.com/deep-learning-digest/lasot-large-scale-dataset-for-object-tracking-models-a4ff0eb03611)) includes [this figure](https://miro.medium.com/max/1050/1*B7Ia6c7nypK3pS5XKSbq6Q.png) from that dataset's paper [1]. That plot shows the number of videos and average number of frames for standard datasets. The product of these factors for GOT-10K (1,500,000 = 10,000 * 150) and LaSOT (3,875,000 = 1,550 * 2,500) are similar in scale to the 64- and 128-frame *PathTracker* datasets (1,280,000 = 20,000 * 64 and 2,560,000 = 20,000 * 128). We will mention this similarity in the SI of our paper. TrackingNet is indeed bigger. Our InT+TransT is the state of the art on that dataset.
> > >
> > > [1] Fan H. et al., 2020.  LaSOT: A high-quality benchmark for large-scale single object tracking. International Journal of Computer Vision.

---

### Author Response · Authors · 2021-08-10
**Response to all reviewers. Thank you!**

We thank the reviewers for their efforts and invaluable suggestions. The reviewers indicated that the paper would benefit from the following additions:
1. A more extensive analysis of why state-of-the-art (SOTA) video analysis models like the R3D and Timesformer fail to solve long *PathTracker* videos (**utTc**).
2. Additional control experiments for the InT circuit model on the real-world tracking challenges (**utTc**).
3. A Kalman Filter baseline on *PathTracker* (**JexJ**).
4. An extended discussion of Multi-Object Tracking and older object tracking models in computer vision (**JexJ** & **hxQK**).

We have performed new experiments to address the first three issues.
1. We gathered new human psychophysics data on the short [32-frame] *PathTracker* that SOTA models were able to solve, and found that these models explain significantly less variance in human decision making on this dataset than our Index-and-track (InT) model, Evidently SOTA video analysis models are biased to learn fundamentally different strategies than humans. While this is not causal evidence that the difference between humans and video analysis models is necessarily linked to its failure to solve *PathTracker*, it is a noteworthy correlation.
2. Our InT+TransT model, which is the state-of-the-art in object tracking, outperforms a control model in which we replace the InT with a ConvGRU.  InT+TransT solves *PathTracker* **and** is SOTA in tracking objects in the real world.
3. A Kalman Filter is unable to track the target object in *PathTracker*.
4. Finally, in response to **JexJ** and **hxQK**, we will integrate our discussion of Multi-Object Tracking that is currently in the Supplementary Materials into the main text and include the suggested citations.

Our paper makes several significant contributions to vision science, cognitive science, and computer vision. The new experiments requested by the reviewers greatly strengthen our work. We look forward to the discussion period for any additional feedback the reviewers might have.

---

> ### Author Response · Authors · 2021-08-16
> **Looking forward to discussions with the reviewers!**
>
> Dear Reviewers,
>
> We hope that you will have a chance to check out our responses soon. Please let us know if there are any more points of clarification needed and if you have any remaining concerns about the paper.
>
> Best,
> The authors

---

> > ### Author Response · Authors · 2021-08-26
> > **Thank you**
> >
> > Dear Reviewers,
> >
> > Thank you again for your time and effort! Please let us know if you have any more points of clarification before the reviewing period ends next week.
> >
> > Best,
> > The authors

---

> > > ### Comment · Reviewer_utTc · 2021-08-30
> > > **Strong responses to reviewers**
> > >
> > > Thank you for your responses to reviewer comments. They addressed my (relatively minor) concerns and the inclusion of various Kalman-filter-based baselines makes the results much stronger + connects it better to classical work in MOT.
> > >
> > > Given that your novel algorithm (a) can solve tasks that supposedly more general ANNs cannot, (b) does so in a way more consistent with how humans do it, and (c) improves on SOTA tracking in the wild, I think this work makes a meaningful contribution to the field of ANN-based object tracking. Although this is not my specialty, and I am glad that the other reviewers suggested baselines and tasks that are well-established within the tracking subfield, I think the NeurIPS community would greatly benefit from knowing about the limitations of standard large ANNs on forms of tracking where humans excel.

---

### Decision · Program_Chairs · 2021-09-27

**Decision:**

Accept (Poster)

**Comment:**

Having read the paper carefully and looked at the reviews, I am pretty convinced this is a useful contribution and should definitely be accepted.  Two of the reviewers gave the paper high scores -- but not a free pass, having asked detailed questions and gotten apparently satisfying responses from the authors. I see that one of the reviewers is very skeptical, and remains unsatisfied by the authors' responses.  But looking at the author responses, and having tried to probe that reviewer's thoughts myself, it seems to me that the reviewer's skepticism is rather biased (e.g. not responding to logical questions, but just kind of baked in).  So I've decided to largely discount that low reviewer's comments in making my final decision.